# Large anomalies in future extreme precipitation sensitivity driven by atmospheric dynamics

Lei Gu[1,2], Jiabo Yin [1]✉, Pierre Gentine [3,4], Hui-Min Wang [5], Louise J. Slater [6], Sylvia C. Sullivan [7], Jie Chen[1], Jakob Zscheischler [8] & Shenglian Guo[1]

Increasing atmospheric moisture content is expected to intensify precipitation extremes under climate warming. However, extreme precipitation sensitivity (EPS) to temperature is complicated by the presence of reduced or hook-shaped scaling, and the underlying physical mechanisms remain unclear. Here, by using atmospheric reanalysis and climate model projections, we propose a physical decomposition of EPS into thermodynamic and dynamic components (i.e., the effects of atmospheric moisture and vertical ascent velocity) at a global scale in both historical and future climates. Unlike previous expectations, we find that thermodynamics do not always contribute to precipitation intensification, with the lapse rate effect and the pressure component partly offsetting positive EPS. Large anomalies in future EPS projections (with lower and upper quartiles of −1.9%/°C and 8.0%/°C) are caused by changes in updraft strength (i.e., the dynamic component), with a contrast of positive anomalies over oceans and negative anomalies over land areas. These findings reveal counteracting effects of atmospheric thermodynamics and dynamics on EPS, and underscore the importance of understanding precipitation extremes by decomposing thermodynamic effects into more detailed terms.

Anthropogenic greenhouse gas concentrations have risen sharply since the second industrial revolution, warming the atmosphere[1–3] and resulting in precipitation intensification in several regions across the globe[4–7]. The sensitivity of global mean precipitation to anthropogenic warming (about 2–3%/°C) is regulated by the atmospheric energy budget between condensation heating and radiative cooling[8]. Extreme precipitation ($P_e$), which is less constrained by energetic limitations, is usually more sensitive to atmospheric warming than mean precipitation, and may exacerbate hydrological extremes such as floods and debris flows worldwide[9–11]. Increases in heavy precipitation are

expected to challenge the current design of flood protection measures and the implementation of risk-management strategies, damaging roads, power grids and other infrastructure and environmental systems[12]. There is thus a pressing need to understand the physical mechanisms behind the sensitivity of $P_e$ to both natural and anthropogenic climate changes.

Following the Clausius-Clapeyron (CC) relationship, the atmospheric moisture holding capacity should increase with warming temperatures at a rate of ~7%/°C. This scaling has been regarded as an important starting point for projecting $P_e$. However, a large and

[1]State Key Laboratory of Water Resources Engineering and Management, Wuhan University, Wuhan, Hubei 430072, P.R. China. [2]Hubei Key Laboratory of Digital River Basin Science and Technology, Huazhong University of Science and Technology, Wuhan 430074, China. [3]Department of Earth and Environmental Engineering, Columbia University, New York, NY, USA. [4]Climate School, Columbia University, New York, NY, USA. [5]Department of Civil and Environmental Engineering, National University of Singapore, Singapore, Singapore. [6]School of Geography and the Environment, University of Oxford, Oxford, UK. [7]Department of Chemical & Environmental Engineering, University of Arizona, Tucson, AZ, USA. [8]Department of Computational Hydrosystems, Helmholtz Centre for Environmental Research, Leipzig, Germany. ✉e-mail: jboyn@whu.edu.cn

growing body of evidence reports divergent sensitivities of $P_e$ to near-surface temperatures ($T$), varying from super CC-scaling rates (i.e., >7%/°C) to decreases with rising temperatures[13–16]. The reason for this divergence is that CC scaling only tells part of the story: $P_e$ is also a function of local vertical motion (atmospheric dynamics) and available atmospheric moisture (thermodynamics)[17–21]. Variations in large-scale atmospheric circulation and local weather patterns can also contribute to deviations from CC scaling[22–24]. For instance, enhanced vertical velocity associated with deep convection in the tropics or extratropical cyclones may intensify $P_e$ and lead to super CC rates[25–27]. Thermodynamic factors such as a less steep moist-adiabatic lapse rate with warming can also decrease the vertically integrated saturation specific humidity and thus weaken precipitation sensitivity[28,29].

Different physical processes are involved in precipitation generation in diverse geographical areas. Three EPS (i.e., $P_e$-$T$ scaling relationship) regimes have been widely reported in the literature. Monotonically increasing EPS is usually found in high latitudes, while the tropics are dominated by monotonically decreasing scaling[18]. Over most regions of the globe, both observational records and model simulations exhibit a "hook-like" structure, in which precipitation intensity generally increases with warming but decreases beyond a peak-point temperature ($T_{pp}$)[30,31]. More recently, it has been found that the EPS is not stationary but can shift along wetter and warmer directions in future climates[32,33]. However, the physical mechanisms underpinning the contribution of atmospheric thermodynamics and dynamics to EPS, as well as the shifting pattern of thermodynamic versus dynamic effects under climate warming, remain unexamined.

Here, we decompose the dynamic and thermodynamic components of EPS to explore the underlying physical mechanisms behind EPS and its shifting patterns under climate change. First, we employ a physical diagnostic scaling approach to detect three EPS regimes using ERA5 reanalysis data (20 pressure levels) and climate simulations from the latest Coupled Model Inter-comparison Project phase 6 (CMIP6, all available runs with vertical velocity and temperature profiles at 19 pressure levels; Supplementary Table 1) during both reference (1985–2014) and future (2071–2100) climates. Then, the three EPS regimes are attributed to one dynamic component (represented by changes in vertical velocity, $\omega$) and three thermodynamic components, i.e., the pressure ($pPR$), temperature ($pT$) and lapse rate ($pLR$) components. Moreover, we project future shifts in the three regimes under climate change (i.e., we diagnose EPS anomalies by comparing future EPS relative to historical EPS) and disentangle the drivers of future EPS anomalies within the CMIP6 experiments. This study represents, to our knowledge, the detailed understanding of the physical mechanisms responsible for changing EPS regimes in a warming world.

## Results
### Physical diagnostic of the ERA5 reanalysis data and CMIP6 experiments
We first explore the extent to which our physical decomposition diagnostic approach accurately describes $P_e$, i.e., the 99th percentile of daily precipitation above 0.1 mm/day. The evaluation is performed by using the ERA5 reanalysis data and CMIP6 climate experiments (more details in the "Methods" section):[15]

$$P_e \sim -\left\{ \omega \frac{dq_s}{dp} \bigg|_{\theta^*} \right\} \qquad (1)$$

where $P_e$ is estimated as the mass-weighted vertical integral of a condensation rate. This condensation rate is the product of the vertical velocity ($\omega$) and the gradient of the saturation specific humidity ($q_s$) at a constant saturation equivalent potential temperature ($\theta^*$). We use daily mean surface pressure, vertical velocity and temperature on the day of the $P_e$ from ERA5 (20 pressure levels) and CMIP6 outputs (19 pressure levels) to estimate the right-hand side of Eq. (1). As the

column-integrated net condensation is usually used to reflect extreme precipitation at the daily time scale[15], the changes in precipitation efficiency are neglected here.

The estimated $P_e$ from this physically based diagnostic accurately reproduces the spatial pattern of $P_e$ in ERA5, with a spatial correlation coefficient of 0.95 ($p < 0.001$) over the globe and deviations within ±10 mm/day in ~92% of global areas (the fractional area is calculated by considering weights in different latitudes; Fig. 1a, b). However, the physical diagnostic underestimates $P_e$ in a few regions such as South America, Sub-Saharan Africa, and the eastern U.S.A. (Fig. 1b). This underestimation may be due to the omission of other factors, such as the effects of topography and of precipitation efficiency, which both relate to microphysical impacts on $P_e$[21]. The diagnostic performs better within CMIP6 than ERA5, with a spatial correlation coefficient of 0.99 ($p < 0.001$) in both historical (1985–2014) and future (2071–2100) climates (Fig. 1c–f).

We then evaluate the performance of this diagnostic approach in detecting the EPS regimes (Supplementary Fig. 1). The actual (ERA5 and raw model outputs, Supplementary Fig. 1d–f) and diagnostic-based (Supplementary Fig. 1j–l) EPS rates exhibit some differences in the decreasing branch of the hook structure, with the diagnostic overestimating the negative slope. For instance, the spatially averaged actual rates (−9.9%/°C) are lower in magnitude than the diagnostic-based EPS rates (−15.3%/°C) in ERA5. The overestimation is alleviated in the CMIP6 simulations, with an average actual rate of −10.1%/°C (−9.5%/°C) and diagnostic based rate of −10.3%/°C (−9.7%/°C) in the historical (future) climate experiments (Supplementary Fig. 1). The spatial correlation coefficients are high for both the ERA5 (>0.6, $p < 0.001$) and CMIP6 (>0.81, $p < 0.001$) datasets. These results broadly confirm that the physical diagnostic approach can effectively reproduce the EPS, providing high confidence in constraining the main drivers of EPS regimes.

### Decomposition of thermodynamic and dynamic contributions
The varying EPS regimes (i.e., monotonically increasing, hook-like and monotonically decreasing structures) motivate us to disentangle the thermodynamic and dynamic contributions to changes in $P_e$. We first examine the EPS regimes under the total forcing using the ERA5 reanalysis and CMIP6 experiments in the historical climate (Figs. 2–3). The widespread presence of a $T_{pp}$ (Fig. 3a, c) reveals the dominance of a hook structure (an increase followed by a decrease of $P_e$ with rising temperature) in constraining EPS (over 65.3% of the globe in ERA5 and 63.9% in CMIP6), and partly due to moisture limitation in warmer environments[31–34]. Strong positive scaling rates (higher than 15%/°C) are mainly observed over the subtropical oceans, while the mid-to-high latitude regions exhibit near- or super-CC scaling rates in the ascending branch of a hook structure (Fig. 2a, c). By fitting regressions to the decreasing branch (beyond $T_{pp}$) of the hook structure, we find the estimated scaling rates in the subtropics are negative and lower than those found in the mid-to-high latitudes (Fig. 2b, d). A monotonically increasing scaling with near-CC rate typically prevails in high-latitude oceans (18.0% of the globe in ERA5, and 23.2% in CMIP6). Negative scaling mainly emerges in the tropics, accounting for 16.7% (12.9%) of the globe in ERA5 (CMIP6 experiments), with scaling rates varying from −15 to −9%/°C (Fig. 2a, c). We further explore the robustness of the binning scaling method in detecting EPS by using a quantile-regression based technique instead[35,36]. The results are very similar: most mid-latitudes show a hook-like structure, low latitudinal regions present a negative scaling and high latitudes are dominated by monotonically increasing scaling (Supplementary Fig. 2).

As $P_e$ is a function of available atmospheric moisture (thermodynamics) and local vertical motion (atmospheric dynamics)[17–20], the EPS regime should also be governed by thermodynamic and dynamic effects. When only focusing on the thermodynamic components of EPS regimes, in which temporal variations of the vertical velocities are

neglected, the globe consistently exhibits a monotonically increasing scaling behaviour (Fig. 3b, d), suggesting that the negative scaling in high temperatures is driven by atmospheric dynamics. To elucidate the physical mechanisms behind these complex EPS regimes, we further propose a detailed decomposition of these thermodynamic components. In terms of thermodynamic controls, $P_e$ depends on the moist-adiabatic pressure slope of the saturation specific humidity ($s = \frac{dq_s}{dp}\big|_{\theta^*}$). As $s$ is the sum with respect to the partial derivative of pressure in saturation specific humidity ($\frac{\partial q_s}{\partial p}$) and the product of the gradient with respect to temperature in saturation specific humidity and the lapse rate ($\frac{\partial q_s}{\partial T} \cdot \frac{dT}{dp}\big|_{\theta^*}$)[37], the thermodynamic component can be partitioned as follows (see "Methods" for details):

$$
\begin{aligned}
P_{the} \sim & -\left\{\omega_{avg}\frac{\partial q_s}{\partial p}\right\} - \left\{\omega_{avg}\frac{\partial q_s}{\partial T}\cdot\left(\frac{dT}{dp}\Big|_{\theta^*}\right)_{avg}\right\} \\
& + \left(-\left\{\omega_{avg}\frac{\partial q_s}{\partial T}\cdot\frac{dT}{dp}\Big|_{\theta^*}\right\} + \left\{\omega_{avg}\frac{\partial q_s}{\partial T}\cdot\left(\frac{dT}{dp}\Big|_{\theta^*}\right)_{avg}\right\}\right) P_{the} \sim pPR + pT + pLR
\end{aligned}
\tag{2}
$$

where $P_{the}$ denotes extreme precipitation forcing only by thermodynamics; $p$ and $(\frac{dT}{dp}\big|_{\theta^*})_{avg}$ indicate pressure and temporal average lapse rate, respectively.

With the decomposition in Eq. (2), we can now quantify the relative contributions of $pT$, $pPR$, and $pLR$ to EPS. The total thermodynamic scaling rates range between 1%/°C and 8%/°C in most areas of the globe (Supplementary Fig. 3). The $pT$ term, i.e., the CC scaling, strengthens $P_e$ with warming, as supported by the monotonic and positive scaling relationships (Supplementary Figs. 4c, 5c). This term also dominates, showing the largest contribution to EPS over 48.5% of the globe (Fig. 3e). However, the lapse rate effect (i.e., $pLR$) offsets this positive scaling, particularly over mid-to-high latitude oceans (Supplementary Figs. 4d, 5d), where the less steep lapse rate with warming weakens the ascent rate and leads to deviations from the CC relationship. The positive thermodynamic scaling is also reduced by $pPR$, the integral of saturation specific humidity dependence on each pressure level, even though it accounts for a relatively small contribution to EPS (Supplementary Figs. 4b, 5b). Overall, the thermodynamic components do not always intensify extreme precipitation. The partial offset of CC scaling by the lapse rate effect and the pressure component means that the thermodynamic effect should be decomposed into more detailed components.

We next evaluate the contribution of the dynamic effect to EPS by subtracting the thermodynamic scaling rates (at temporally fixed $\omega$)

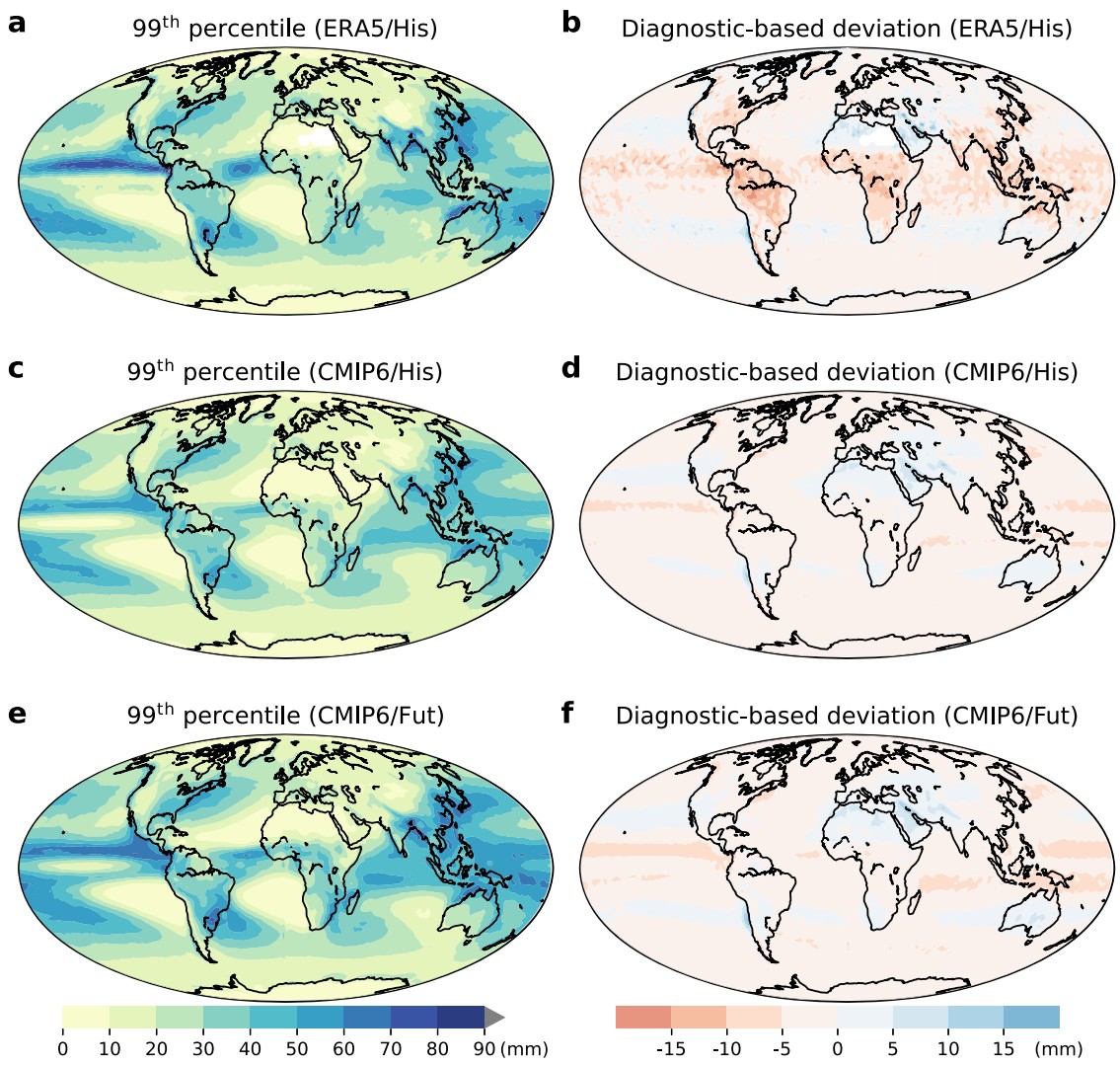

**Fig. 1 | Consistency of spatial patterns in actual and diagnostic-based daily precipitation extremes. a, c, e** 99th-percentile precipitation based on ERA5 reanalysis (**a**) and CMIP6 multi-model ensemble mean (**c, e**). **b, d, f** Deviation between the physical diagnostic and actual 99th-percentile precipitation based on ERA5 reanalysis (**b**) and CMIP6 multi-model ensemble mean (**d, f**). The precipitation extremes are estimated during the 1985–2014 (His) and 2071–2100 (Fut) periods, respectively.

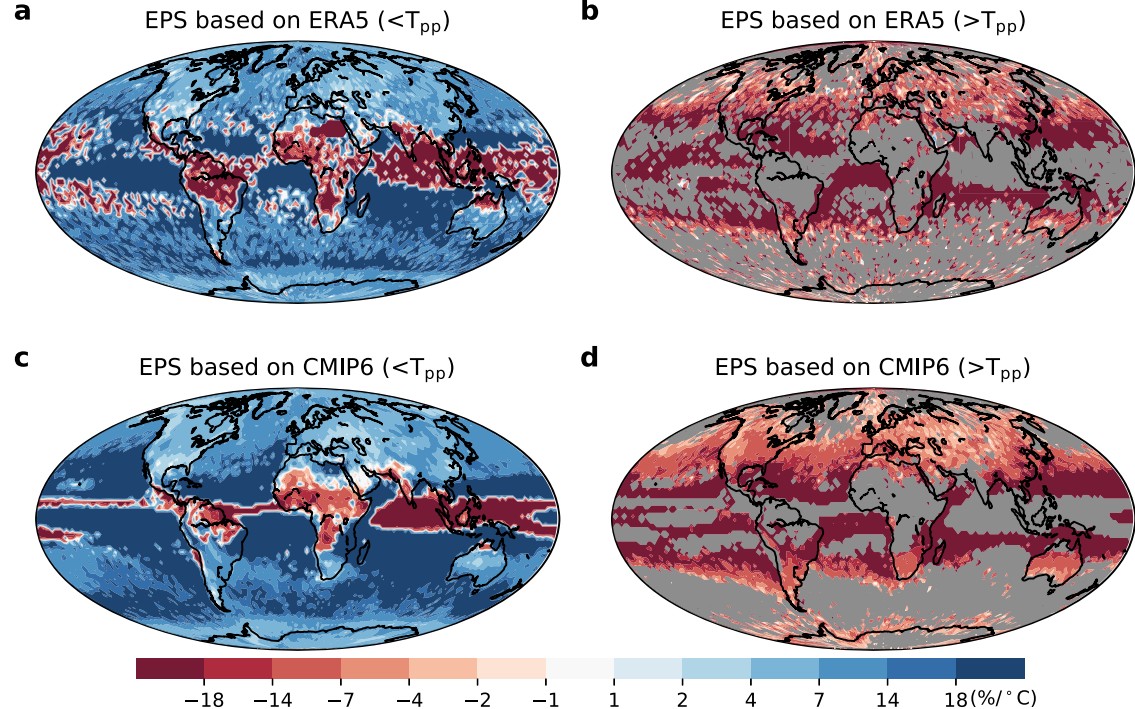

**a** EPS based on ERA5 (<$T_{pp}$)

**b** EPS based on ERA5 (>$T_{pp}$)

**c** EPS based on CMIP6 (<$T_{pp}$)

**d** EPS based on CMIP6 (>$T_{pp}$)

−18 −14 −7 −4 −2 −1 1 2 4 7 14 18 (%/°C)

**Fig. 2 | Extreme precipitation sensitivity (EPS) based on ERA5 and CMIP6 during the reference 1985–2014 period. a, b** 99th-percentile precipitation-temperature scaling rate based on ERA5 reanalysis before $T_{pp}$ (<$T_{pp}$, exhibiting three EPS regimes) and after the $T_{pp}$ (>$T_{pp}$, i.e., only the decreasing branch in the hook-like scaling). **c, d** Results based on CMIP6 average multi-model ensemble experiments. Monotonic scaling types (monotonically increasing and decreasing regimes) in (**b, d**) are masked in grey.

from the total scaling (Supplementary Figs. 3–5). By comparing with the thermodynamic scaling, the dynamic scaling shows larger spatial variability and dominates the EPS in low-to-mid latitude regions (accounting for 51.5% of the globe; Fig. 3e, f). More specifically, the dynamic scaling ranges from extremely negative rates (<−15%/°C) across the Intertropical Convergence Zone (ITCZ)[34] to strongly positive scaling rates (>15%/°C) across the subtropics, followed by <7%/°C scaling rates in the mid-to-high latitudes (Supplementary Fig. 3). The dynamic effect on EPS (Supplementary Figs. 4–5) is predominantly induced by variations in $\omega$. Over the ITCZ, as temperature rises, reduced $\omega$ suppresses the development of deep convective systems[32], thus resulting in negative EPS. In mid-to-high latitudes, when temperatures are below $T_{pp}$, $\omega$ tends to increase with rising temperatures, thus enhancing vertical updraft and moisture availability, and thereby promoting $P_e$. When local temperatures exceed the $T_{pp}$, reduced $\omega$ constrains moisture vertical transport and thus inhibits $P_e$. As a result, most global areas usually generate a hook structure, particularly in Australia, North America, Europe, Asia, and the Atlantic and Pacific Oceans.

We also perform the decomposition at the hourly scale and for the extended warm season (May-October in the Northern Hemisphere and November-April in the Southern Hemisphere; Fig. 4). Based on the ERA5 dataset, we find the hook-like structure still governs global EPS over 68.4% of the globe at the hourly time scale and 59.4% during the warm season. The total scaling rates remain almost unchanged when comparing hourly and daily time scales, and the results are robust when we focus on warm season in the mid-to-high latitudes (>30°N and <30°S). This robustness can be attributed to the fact the the $pT$ term (see Eq. (10)) dominates the total scaling over these regions and it is stable regardless of different time scales (seasons). In contrast, in the tropical regions between ~30°S and ~30°N where the dynamic term prevails, the total scaling varies across different temporal scales and seasons. This partly reflects the high sensitivity of the dynamic term to rising temperature.

## Shifting thermodynamic and dynamic controls under climate change

To disentangle whether and how the thermodynamic *versus* dynamic controls might evolve with climate change, we examine possible future shifts in EPS based on the Shared Socioeconomic Pathway 5-8.5 (pessimistic scenario) within the CMIP6 experiments (Fig. 5). For the total forcing (Fig. 5a), the scaling types remain nearly unchanged over most areas of the globe in the future climate (2071–2100). However, the hook-dominated regions show a noticeable increase in $T_{pp}$ (even higher than 10.0 °C for some regions), indicating a future shift in the hook structure towards warmer and wetter conditions (Fig. 5i). However, these changes in $T_{pp}$ are slower than local warming rates (defined as future minus reference local mean temperature, $T_{as}$), with global spatially averaged increases of 3.5 °C and 4.6 °C, respectively (Supplementary Figs. 6–7). If $T_{as}$ does indeed exceed $T_{pp}$, the EPS regime may shift to a descending scaling, potentially mitigating future extreme precipitation intensification. However, $T_{as}$ remains substantially lower than $T_{pp}$ in both reference and future climates across most regions of the globe (Supplementary Figs. 8–9), even though the difference between $T_{as}$ and $T_{pp}$ might shrink due to faster increases in $T_{as}$ under future climate warming. Most regions still exhibit positive EPS scaling as a result, and future precipitation is likely to intensify by the end of 21st century.

We then investigate the spatial distribution of EPS anomalies (i.e., future minus reference scaling rates) within CMIP6 projections. In the regions which exhibit a monotonically decreasing scaling in the reference climate, the EPS anomalies are mainly driven by the dynamic component (Fig. 5a–f), with lower and upper quartiles of −1.9%/°C and 8.0%/°C, respectively (Fig. 5a, d). In the hook-dominated regions, the lower and upper quartiles of the EPS anomalies are −1.5%/°C and 3.1%/°C (Fig. 5a, d). Large increases in scaling rates of the ascending branch of the hook structure (i.e., below the $T_{pp}$) are projected over the subtropical oceans (Fig. 5a). In contrast, the negative scaling of the

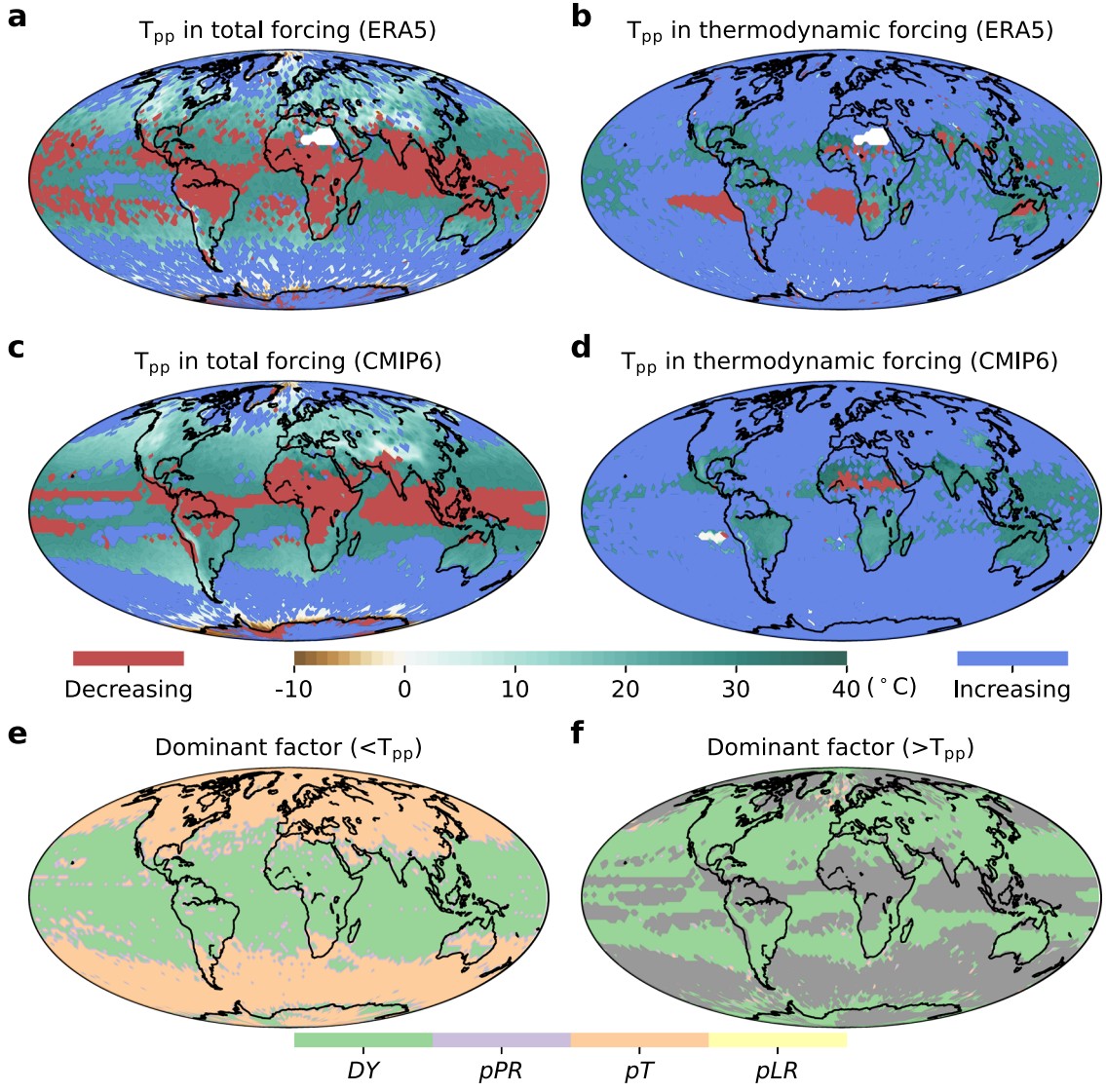

**Fig. 3 | The peak-point temperature ($T_{pp}$) in total and thermodynamic forcing as well as the dominant factor contributing to extreme precipitation sensitivity within the reference climate. a–d** Total and thermodynamic forcing based on ERA5 (**a, b**) and CMIP6 (**c, d**) within the 1985–2014 period, respectively. **e, f** The dominant factor with maximum contribution to EPS before $T_{pp}$ (<$T_{pp}$, showing three EPS regimes) and after $T_{pp}$ (>$T_{pp}$, only the additionally decreasing branch in hook-like scaling) within CMIP6. *DY, pPR, pT* and *pLR* represent dynamic and thermodynamic pressure, temperature and lapse rate components, respectively. Monotonic scaling type in (**f**) is masked in grey.

descending branch (i.e., above the $T_{pp}$) is projected to reduce in magnitude under climate change (Fig. 5d). The dynamic component still plays a primary role in driving these scaling rates anomalies (Fig. 5c, f), in spite of minor changes also result from the thermodynamic scaling (Fig. 5b, e). At higher latitudes, the thermodynamic components become increasingly important in modulating EPS, with lower and upper quartiles of EPS anomalies of −0.2%/°C and 3.0%/°C (Fig. 5b).

As different thermodynamic components might have divergent effects on EPS anomalies, we further estimate the contribution of all thermodynamic terms (i.e., *pPR, pT, pLR*) and dynamics (*DY*) over the globe, land and oceans, respectively (Fig. 6 and Supplementary Figs. 10–11). Large positive zonal average EPS anomalies mainly occur between ~30°S and ~30°N (Fig. 6a), driven by the positive EPS anomalies over oceans. The land areas consistently demonstrate negative EPS anomalies in the latitude bands (Fig. 6f, k). These large land-ocean discrepancies in EPS anomalies can mainly be attributed to moisture limitation over land. In the context of a future warming climate, extreme precipitation is projected to increase sharply over

the oceans with rising temperature, given sufficient moisture supply. Over land areas, although saturation vapour pressure still strongly increases with warming, enhanced vapour pressure deficit results from moisture limitation can limit extreme precipitation intensification.

From the decomposition, the dynamic component explains the large anomalies at low latitudes (Fig. 6b, g, l). Indeed, it controls EPS anomalies across ~80.9% of the globe (Fig. 5g, h) and is larger than any of the thermodynamic components. Specifically, $\omega$-related mechanisms including the strengths of circulation are highly sensitive to a warming climate, and thus largely altering EPS. The EPS anomalies slowly weaken with the increase in latitude and then suddenly strengthen over the oceans at ~60°S and ~60°N (Fig. 6k, n). This sharp increase in oceanic EPS anomalies is mainly driven by the *pT*. The *pLR* and *pPR* components remain almost unchanged with climate warming and have more limited impacts on the EPS anomalies. These results further emphasize the necessity of decomposing thermodynamic effects into different terms due to their opposing effects.

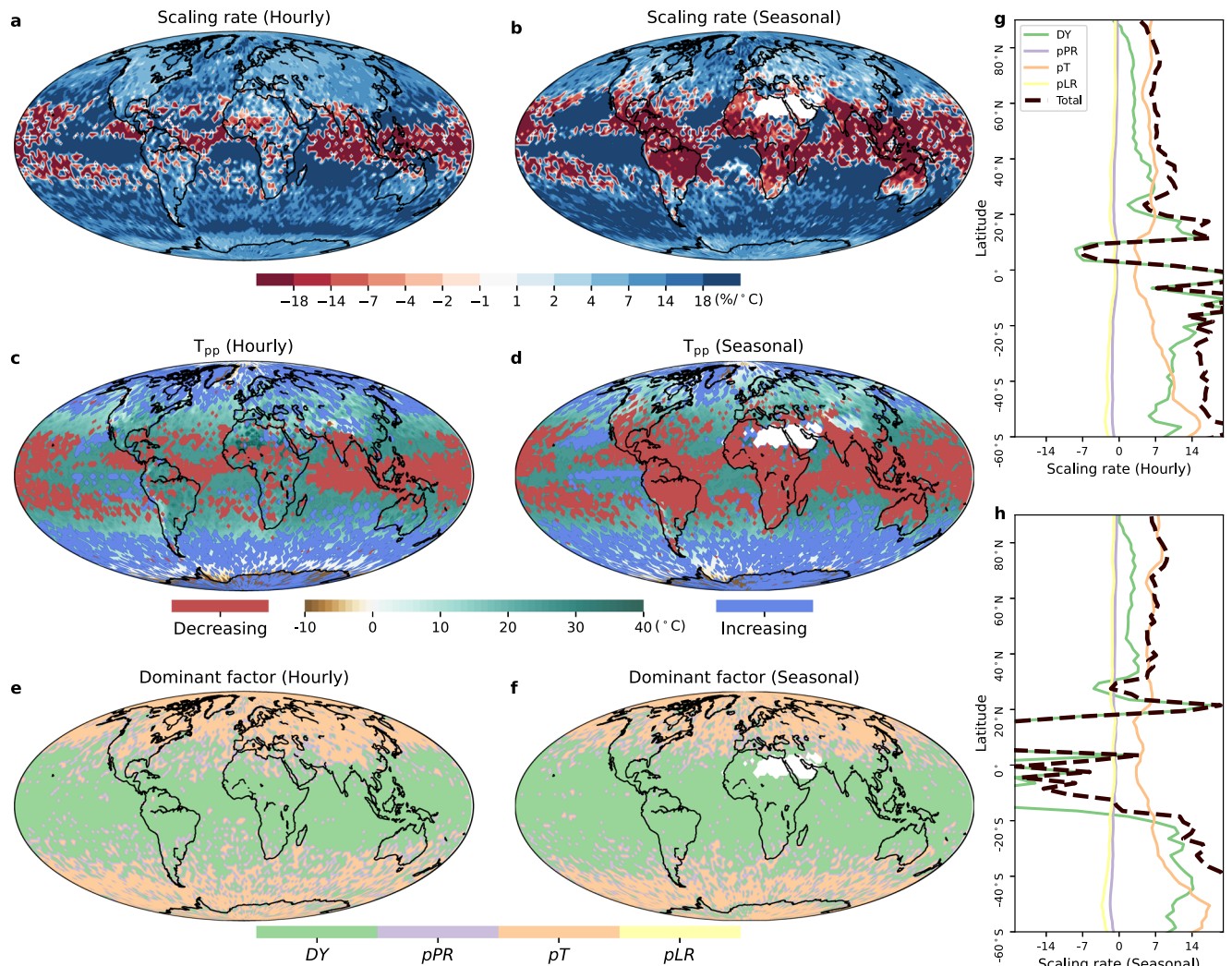

**Fig. 4 | Extreme precipitation sensitivity, the peak-point temperature ($T_{pp}$) and dominant factor of the extreme precipitation-temperature ($P_e$-$T$) relationship during the reference 1985–2014 period from the ERA5 dataset. a, c, e** Scaling rate, $T_{pp}$ and dominant factor of the $P_e$-$T$ relationship during the reference period at the hourly scale. **b, d, f** Scaling rate, $T_{pp}$ and dominant factor of the $P_e$-$T$ relationship during the reference period for the warm season (May to October in the Northern Hemisphere and November to April in the Southern Hemisphere). **g** Zonal total scaling rate and the *DY, pPR, pT* and *pLR* contributions at the hourly scale. **h** Zonal total scaling rate and the *DY, pPR, pT* and *pLR* contributions for the warm season.

## Exploring the thermodynamic versus dynamic physics contributing to EPS regimes

We further explore the physical mechanisms behind $P_e$ changes by grouping global grid cells into three EPS regimes under historical and future climates. Under future climates, the scaling rates of the $P_e$-$T$ relationship cannot be simply extrapolated from the historical scaling, but tend to shift toward a warmer and wetter conditions, thereby promoting $P_e$ intensification (Fig. 7a, d, g).

How do the dynamics and thermodynamics control the monotonically decreasing, hook-like and monotonically increasing EPS regimes? Dynamics can determine the type of $P_e$-$T$ scaling, supported by the dependence of vertical velocity on temperature. The $\omega$ follows the shape of the $P_e$-$T$ scaling, peaking at the $T_{pp}$ or monotonically changing with temperature (Fig. 7b, e, h; as indicated by the blue dashed/solid black line located at the boundary of blue/grey range). Quantitatively, dynamics is the dominant $P_e$-$T$ factor in regions with a negative scaling, with a contribution of −11.1%/°C (−11.5%/°C) in the historical (future) period, respectively (Fig. 7a). In a warming climate, weakened circulation constrains $P_e$ intensity, leading to negative $P_e$-$T$ scaling in tropical regions (Fig. 7c). In most global land areas with a hook structure, the *DY* term still plays an important role in shaping EPS

regimes (Fig. 7f). When temperatures are lower than the $T_{pp}$, positive dynamic scaling (2.8%/°C and 3.2%/°C in historical and future climates) due to increased vertical motion enhances $P_e$ (Fig. 7f). These dynamical impacts lead to super-CC scaling rates across historical and future climates. When the environment becomes warmer than $T_{pp}$, reduced vertical velocities (e.g., anticyclonic motion) or subsiding conditions and strong latent heat fluxes and surface cooling synergistically weaken the $P_e$-$T$ scaling[31], counteracting the positive thermodynamic contributions. However, despite the negative or hook-like $P_e$-$T$ relationship, the peak $P_e$ still increases in the future climate (Fig. 7a, d). Over high-latitude oceans, the influence of the dynamic component becomes less important, and precipitation scaling is primarily dominated by the thermodynamics (Fig. 7i).

Thermodynamic components (i.e., *pPR, pT, pLR*) become gradually more important in determining the type of EPS at higher latitudes, with contributions of 1.8%/°C, 3.6%/°C and 4.6%/°C (2.1%/°C, 3.7%/°C and 7.3%/°C) in the three regimes in the historical (future) climate, respectively (Fig. 7a, d, g). Despite the consistently positive overall effect of thermodynamics on the $P_e$-$T$ relationship, the three components show different contribution patterns. Specifically, the *pT* term consistently strengthens $P_e$ intensity, contributing to 3.4%/°C, 5.4%/°C

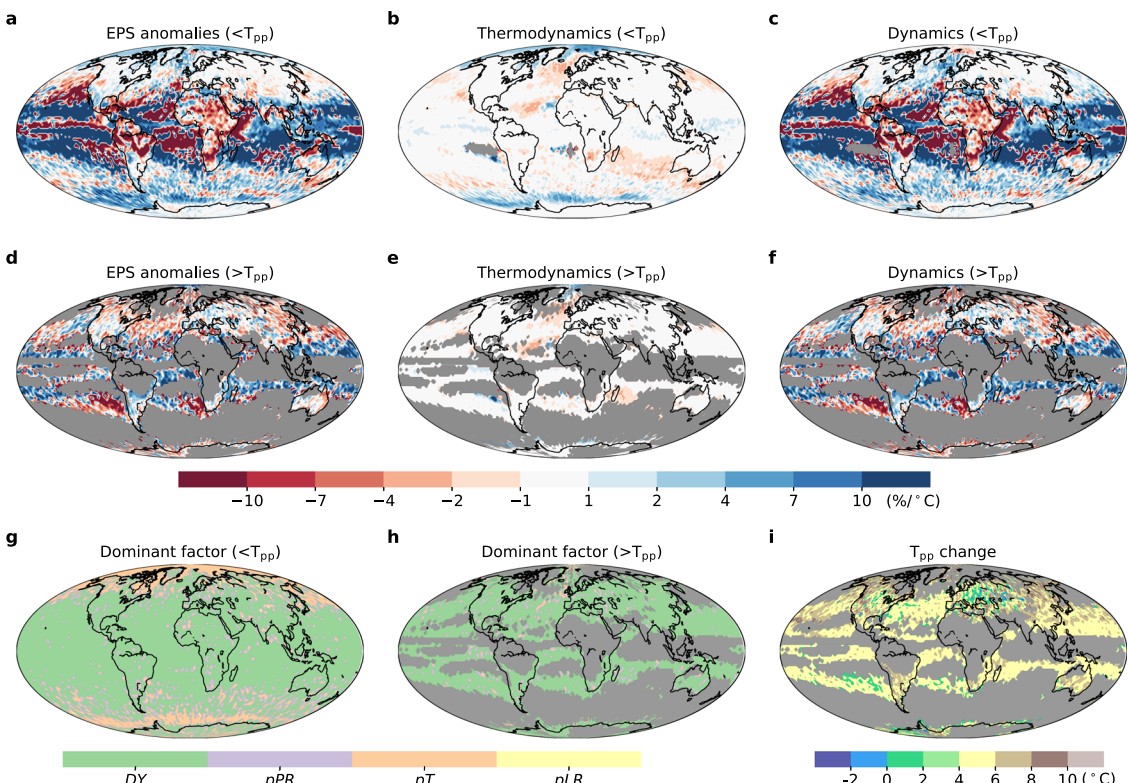

**Fig. 5 | Contribution of thermodynamic versus dynamic components to extreme precipitation sensitivity (EPS) anomalies and the peak-point temperature ($T_{pp}$) changes between the reference and future periods. a, d** EPS anomalies (i.e., relative to the reference period) before ($<T_{pp}$) and after $T_{pp}$ ($>T_{pp}$), respectively. **b, e** Thermodynamic contribution to EPS anomalies before and after $T_{pp}$. **c, f** Dynamic contribution to EPS anomalies before and after $T_{pp}$. **g, h** The dominant factor (showing the greatest contribution among *DY*, *pPR*, *pT* and *pLR* components) contributing to EPS anomalies before and after $T_{pp}$. The monotonically increasing and decreasing regimes (without an additional decreasing branch) are masked in grey in (**d–f, h**). **i** $T_{pp}$ changes projected by CMIP6 multi-model ensemble mean. $T_{pp}$ changes are only presented in the hook-like regime spanning both reference and future periods. Otherwise, in locations corresponding to the monotonically increasing and decreasing regimes, the changing behaviors are masked in grey.

and 6.9%/°C in the three EPS regimes (i.e., monotonically decreasing, hook, and monotonically increasing types), respectively under historical climate. More importantly, the positive scaling rates are projected to increase in a future climate, with rates of 4.0%/°C, 6.0%/°C and 10.8%/°C in the three EPS regimes, respectively. Therefore, intensification of $P_e$ can be attributed to this positive scaling of the *pT* component. In contrast, the *pLR* and *pPR* terms contribute negatively to $P_e$-$T$ scaling, with the *pLR* demonstrating more negative scaling than the *pPR* across three EPS regimes (Fig. 7c, f, i). The lapse rate effect (*pLR*) weakens positive scaling rates by −0.9%/°C, −1.1%/°C and −1.5%/°C (−1.0%/°C, −1.4%/°C and −2.3%/°C) in the three EPS regimes during the historical (future) climate. Meanwhile, the pressure component (*pPR*) also offsets these positive scaling by around −0.7%/°C, −0.7%/°C and −0.8%/°C (−0.8%/°C, −0.9%/°C and −1.3%/°C) during the historical (future) period, respectively.

## Discussion

A physical diagnostic can effectively reproduce the three EPS regimes in both ERA5 reanalysis and CMIP6 simulations and projections. Monotonically decreasing scaling emerges around the ITCZ, monotonically increasing scaling prevails in high latitude oceans, and a hook-like scaling dominates most other regions of the globe. We detect statistically significant $P_e$-$T$ relationships in both ERA5 reanalysis and CMIP6 climate experiments, in line with existing studies[31–33]. As the internal terms of atmospheric thermodynamics demonstrate divergent contributions, we present a detailed decomposition of EPS into one dynamic and three thermodynamic components. Counter to our intuition, we find that the thermodynamic components do not always contribute to precipitation intensification. Although the

thermodynamic temperature (*pT*) term, or CC scaling, strongly enhances EPS, especially in mid-to-high latitudes, the lapse rate term (*pLR*) and the pressure component (*pPR*) can weaken EPS. Specifically, the *pLR* term not only correlates with saturation specific humidity, but is also affected by atmospheric stability and convective[38]. However, these processes are difficult to capture with current convection parameterizations in GCMs, which may result in an underestimation of this term[39]. In addition, we find that the dynamic component varies across different spatial-temporal scales, ranging from negative scaling to more than double CC scaling. We understand that the scaling behaviours cannot be directly applied to predict future precipitation extremes, nor can they be simply extrapolated to project long-term changes in extreme precipitation. However, detailed decomposition of this scaling and unravelling its future shifts could help bound uncertainties in future extreme events and assess how their frequency and intensity will change.

The mechanisms behind extreme precipitation scaling are quite complex in some regions. In tropical regions where EPS is governed by the dynamic term, extreme precipitation is typically associated with storms and cyclones. Other synoptic patterns, including moisture transport from low level jets and upper-level atmospheric rivers, also play a role in modulating EPS[40]. In mid-latitude land regions such as over the Southeast and Midwestern US, Southeast China, and Southern Australia, deep convection dominates extreme precipitation, as indicated by very large convective available potential energy (CAPE) and convective inhibition (CIN) anomalies (Supplementary Fig. 12a–b). This convection is accompanied by high total column water vapour and strong moisture convergence during extreme precipitation (Supplementary Fig 12c–d). Interestingly, these regions all firmly exhibit a

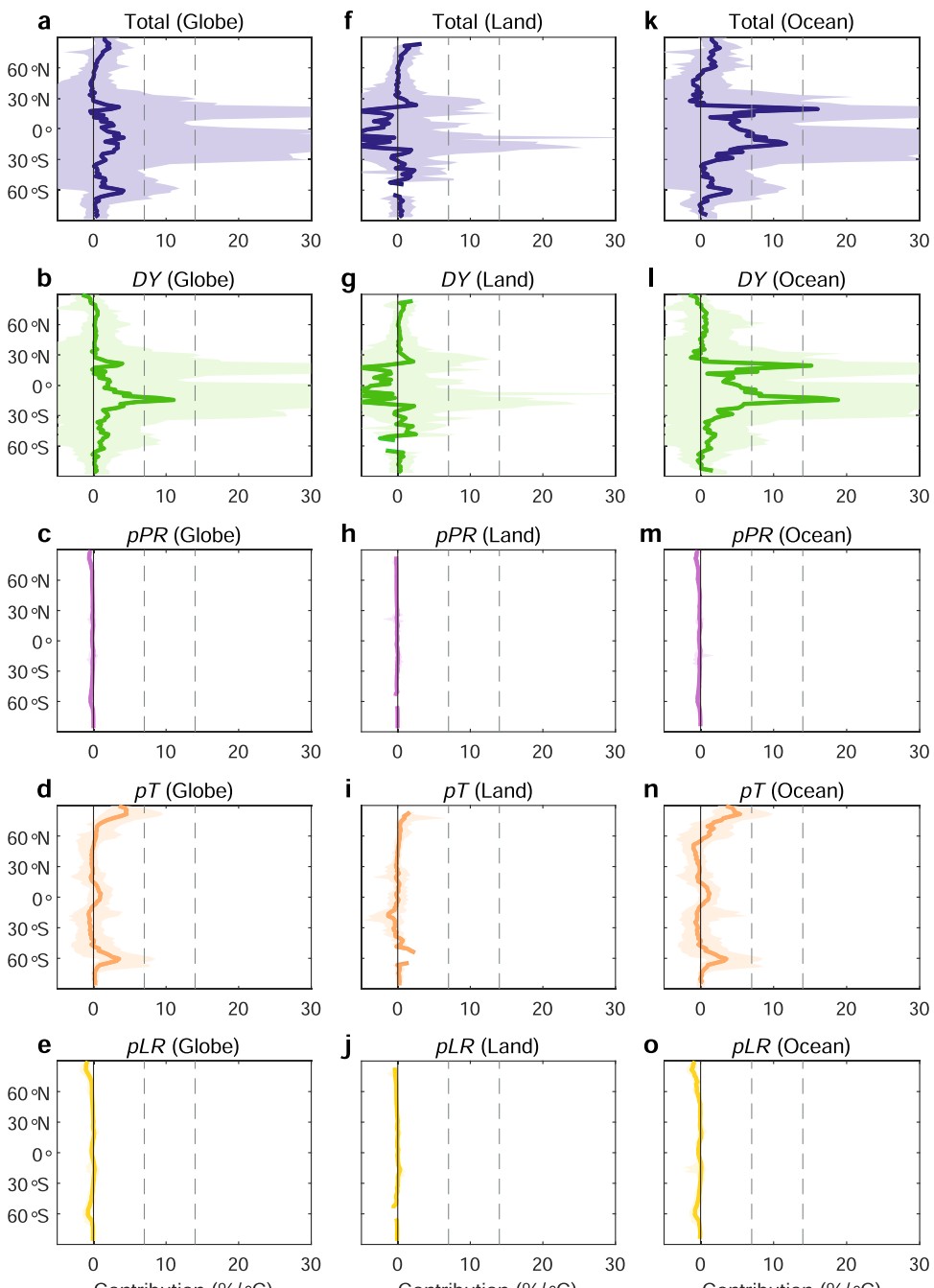

**Fig. 6 | Zonal extreme precipitation sensitivity (EPS) anomalies between the reference and future periods. a** Zonal median EPS anomaly over the globe. **b–e** Contributions of *DY, pPR, pT* and *pLR* to zonal median EPS anomalies over the globe. **f** Zonal median EPS anomalies over global land areas. **g–j** Contributions of *DY, pPR, pT* and *pLR* to zonal median EPS anomalies over global land areas. **k** Zonal median EPS anomalies over global ocean areas. **l–o** Contributions of *DY, pPR, pT* and *pLR* to zonal median EPS anomalies over global ocean areas. The shading shows the range (95th percentiles minus 5th percentiles) of EPS anomalies and each contribution for each latitude. Solid vertical black lines indicate zero scaling, dashed grey lines indicate C-C and double C-C scaling, respectively.

hook-like scaling, using both the binning scaling and quantile regression approaches, at both hourly and daily temporal scales. The physics behind this hook-like structure is multifaceted. Specifically, we find it is determined by the reduced sensitivity of vertical velocity at higher temperatures. Another explanation is the substantial underestimation of convective events at higher temperatures[35,36]. The CAPE corresponding to extreme precipitation is increasing almost monotonically with rising temperature over these regions (Supplementary Fig. 13). However, reduced moisture availability and decreased relative humidity in warmer environments may weaken moisture convergence (Supplementary Fig. 13) and eventually lead to decreasing

precipitation intensity at high temperatures[38]. At higher latitudes which exhibit a monotonically increasing scaling (e.g., in Europe), extreme precipitation is more dependent on low pressure systems and atmospheric rivers than convection and is impacted more by the thermodynamic terms than the dynamic contribution[30]. Current generation of climate models are accompanied by subgrid-scale uncertainties due to their coarse resolution; future studies could combine climate models with machine-learning techniques to further explore the decomposition of EPS at sub-grid cloud-resolving scales[41].

We also decompose the EPS and EPS anomalies using CMIP5 outputs (see Supplementary Table 2) for comparison. In the mid-to-

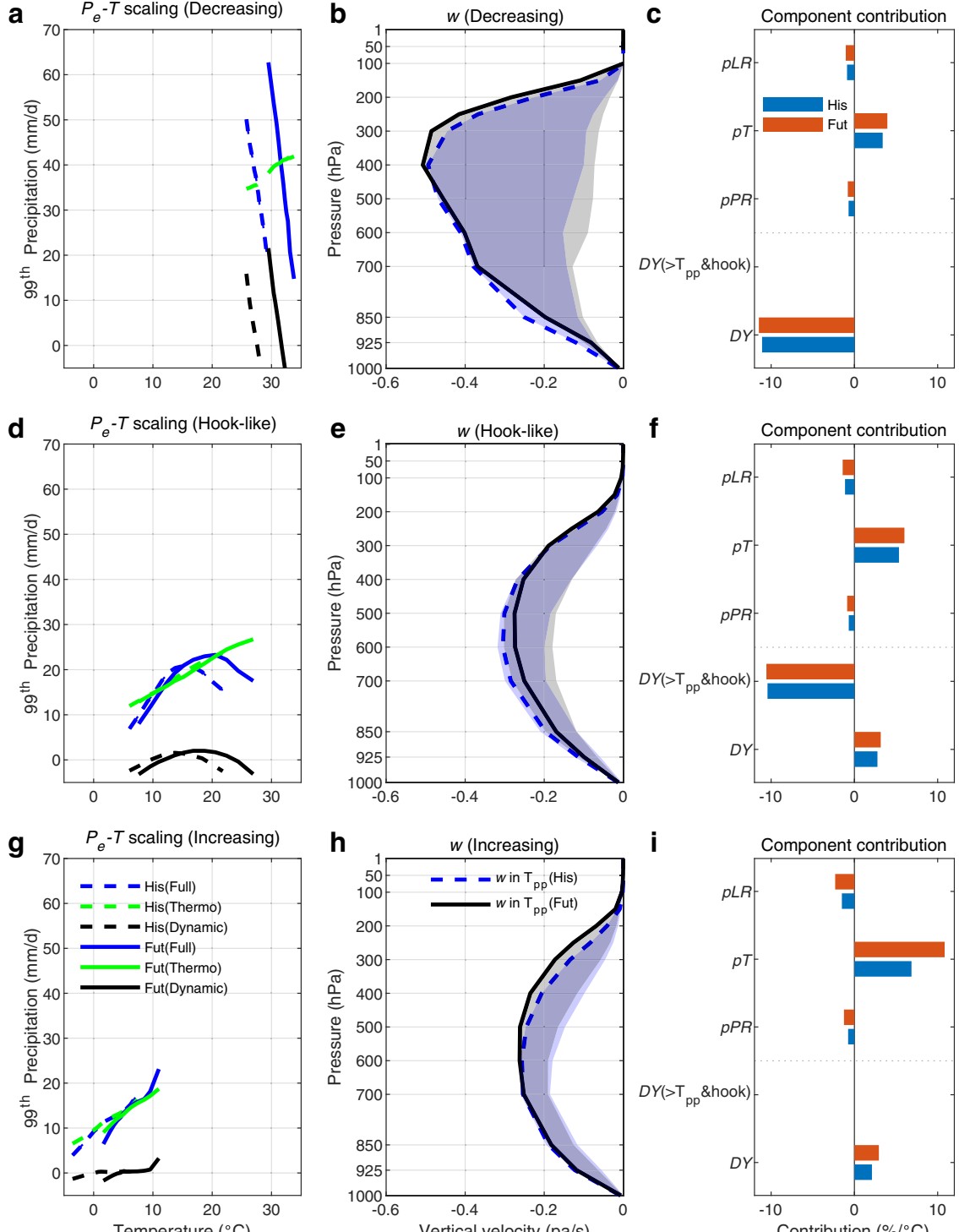

**Fig. 7 | Extreme precipitation-temperature ($P_e$-$T$) scaling and associated component contribution during reference and future periods based on CMIP6 for the three regimes. a**, **d**, **g** Dynamic versus thermodynamic controls on $P_e$-$T$ scaling. **b**, **e**, **h** The vertical velocity ($\omega$) over 12 precipitation-temperature bins. The blue (grey) range is estimated by the minimum and maximum vertical velocity in 12 bins for the historical (His) and future (Fut) periods, whereas the blue dashed (solid black) line indicates the vertical velocity corresponding to the $T_{pp}$. **c**, **f**, **i** The contribution of the one dynamic (*DY*) and the three thermodynamic (*pPR*, *pT*, *pLR*) components to $P_e$-$T$ scaling during the 1985–2014 (His) and 2071–2100 (Fut) period, respectively.

high latitudes, extreme precipitation scaling in the CMIP5 models is similar to the scaling in CMIP6 models during the historical period (Supplementary Fig. 14a–c). The main difference in historical EPS between CMIP5 and CMIP6 lies in the ITCZ region, where, in CMIP5, the strong negative scaling present in ERA5 and CMIP6 disappear. The lack of negative scaling in the tropical region in CMIP5 may be attributed to different parameterization schemes in the models. When comparing future EPS anomalies, the varying regimes across different latitudes found in CMIP6 hold in CMIP5 (Supplementary Fig. 14d–e), although the associated changes in $T_{pp}$ are slightly smaller in CMIP5 than in CMIP6. Overall, most results in CMIP5, including the EPS, EPS anomalies and their decomposition, mirror

those in CMIP6 (Supplementary Fig. 14), adding further credence to our conclusions.

We further estimate the uncertainty of the EPS anomalies using the range of the CMIP6 ensemble. The GCMs project relatively consistent changes in scaling types between the reference and future periods (Supplementary Fig. 15a). Uncertainties in the thermodynamic scaling changes are small; almost all models agree on a consistently positive scaling across the globe, albeit with some small discrepancies in the magnitude (Supplementary Fig. 15b). However, for the hook-dominated regions, projections of changes in $T_{pp}$ vary widely across the ensemble due to uncertainties in the projected dynamic component. The total EPS anomalies associated with the dynamic effects are much more uncertain than the thermodynamic scaling change (Supplementary Fig. 16). This reflects the fact that it is more challenging to project future variations in dynamics[42–44].

Despite the uncertainties in future projections of $P_e$, our analysis provides key physical insights into the shifting behaviour of $P_e$ in a warming climate. Beyond the impact of EPS anomalies, several other possible factors may also alter extreme precipitation intensity, namely precipitation efficiency, cloud microphysics, hydrometeor growth, and atmospheric advection[40,45–47]. As dry-bulb temperature cannot capture humidity-temperature interaction, wet-bulb temperatures may serve as another potential indicator to measure EPS in future work. Synoptic weather events such as cyclones might also affect the $P_e$ and the EPS regimes in the deep tropics[48,49]. Although previous studies[32,33] have shown that the cooling effects of precipitation on near-surface temperatures have little impacts on EPS, this issue still deserves a more systematic investigation. It is noteworthy that the binning method[31] used to associate $P_e$ with $T$ mixes multiple atmospheric processes, and it does not mean changes in extreme precipitation are entirely caused by changing temperature. In reality, the significant relationships between $P_e$ and $T$ quantified by the binning method suggest that changes in extreme precipitation can be reflected in temperature variations. This study takes a step further, revealing the thermodynamic and dynamic mechanisms behind the $P_e$-$T$ relationships.

In conclusion, this work provides the global quantitative assessment of how thermodynamic and dynamic factors regulate $P_e$ shifts in a warming climate. We systematically disentangle the physical mechanisms, in terms of updraft velocity and moist-adiabatic saturation specific humidity, and assess how they may govern future EPS anomalies. We find that the internal thermodynamic components do not always contribute positively to precipitation intensification. Moreover, large EPS anomalies are projected in the future climate, with strongly positive anomalies emerging over oceans and negative anomalies over land areas. These large EPS anomalies are predominantly driven by atmospheric dynamics over ~80.9% of the globe, particularly in tropical and subtropical regions, whereas the thermodynamic effects are much more stable in a continued warming world. Under the impacts of atmospheric dynamics and thermodynamics, extreme precipitation events are projected to continue increasing across the globe, both in terms of their mean and variability (i.e., standard deviation, SD), intensifying future runoff extremes (Supplementary Fig. 17). Our findings suggest an urgent need to increase societal resilience to this changing environment, as precipitation and runoff extremes are likely to intensify in a warming climate, causing major challenges for existing infrastructure and human society.

## Methods
### Reanalysis data
Precipitation, runoff, surface air temperature (2 m) and dew point temperature are obtained from the European Centre for Medium-Range Weather Forecasts (ECMWF) Re-analysis v5 (ERA5)[50]. For the purpose of decomposing precipitation into thermodynamic and

dynamic components, we also use large-scale environmental variables (i.e., sea level pressure, vertical wind velocity and air temperature) at 20 pressure levels ranging between 50 hpa and 1000 hpa (at 50 hpa interval) from ERA5. ERA5 is based on the state-of-the-art Integrated Forecasting System (IFS) Cy41r2 and benefits from the integration of vast amounts of historical observations, new developments in model physics, core dynamics and assimilation techniques, covering a period from 1950 to the present. Specifically, it is a model-based reanalysis product which has uncertainty quantification of varying magnitude across different regions of the globe. We integrate the primary hourly data to a daily temporal and 2° spatial resolution between 1985 and 2014.

### Global climate model data
All available GCM simulations (including 6 models with 7 runs in total) with full surface (e.g., surface temperature, daily precipitation, total runoff, surface relative humidity sea level pressure) and vertical (e.g., vertical pressure velocity and vertical temperature) variables under the historical (1985–2014) and future (2071–2100) periods were retrieved for this study. To better partition the dynamic and three thermodynamic components contributing to precipitation scaling, we use Eday frequency including 19 pressure levels, i.e., 1000, 925, 850, 700, 600, 500, 400, 300, 250, 150, 100, 70, 50, 30, 20, 10, 5, 1 hpa, for vertical profiles. The available GCM runs (main information is presented in Supplementary Table 1) include CanESM5 (*r1i1p2f1*), HadGEM3-GC31-LL (*r1i1p1f3*), INM-CM4-8 (*r1i1p1f1*), INM-CM5-0 (*r1i1p1f1*), MIROC6 (*r1i1p1f1*) and UKESM1-0-LL (*r1i1p1f2* and *r14i1p1f2*). We employ the most pessimistic scenario (i.e., SSP5-8.5) to identify a clear signal of the role of thermodynamic versus dynamic factors in shaping extreme precipitation. The resulting fields are interpolated on a 2° grid using the bilinear method, and we present the multi-model ensemble mean results for the main analysis.

### Definition of extreme precipitation sensitivity
Extreme precipitation sensitivity (EPS) is estimated for each grid cell based on a 'binning-scaling' method[51]. Daily precipitation is 'binned' according to local temperature using 12 bins, in line with previous studies[31–33]. Within each temperature bin (more than 150 precipitation events in each bin), the daily precipitation series is used to estimate the 99th percentile, and the three nearest events to this 99th percentile are averaged to define the daily extreme. In estimating the 99th percentile, we only employ wet days (precipitation >0.1 mm/d) to focus on the intensity rather than the frequency of precipitation, as only the intensity can scale with saturation vapour pressure[52]. The resulting 99th percentile extremes for all 12 temperature bins are smoothed by using a locally weighted regression smoothing method, and the resulting extremes are extracted to characterize the extreme precipitation-temperature relationship. Specifically, three scaling relationships can be identified based on the peak of extreme precipitation and the local temperature at which it peaks ($T_{pp}$). When the $T_{pp}$ is in the lowest (highest) temperature bin, the relationship is defined as having a monotonically decreasing (increasing) scaling structure. Otherwise, the scaling relationship is classified as a hook structure in which precipitation extremes firstly increase with rising temperatures and then decrease at high temperature[33].

The EPS values within each grid cell are then analysed by utilizing the Clausius-Clapeyron relationship, which describes the exponential increase in saturated vapour pressure ($e_s$) with rising temperature:

$$e_s(T) = e_{s0} \exp\left[\frac{L_v}{R_v}\left(\frac{1}{T_0} - \frac{1}{T}\right)\right] \quad (3)$$

where $T_0$ (273.16 K) and $e_{s0}$ (611 Pa) are integration constants, and $L_v$ (2.5 × 10$^6$ J kg$^{-1}$) and $R_v$ (461 J kg$^{-1}$ K$^{-1}$) are latent heat of vaporization

and a vapour gas constant, respectively. For precipitation extremes, the EPS, or the scaling rate ($P_r$) can be estimated as[53]:

$$P_r = \left( \exp^{\frac{\ln P_b - \ln P_a}{T_b - T_a}} - 1 \right) \times 100\% \qquad (4)$$

where $P_a$ and $P_b$ are precipitation extremes for two adjacent temperature bins ($T_a$, $T_b$). $P_r$ can be estimated using least squares linear regression before and after the $T_{pp}$, respectively. In defining the conditional precipitation extremes, the 99th-percentile is based on the wet days (>0.1 mm/day). In addition, we weight each grid cell by its area-average in this study, to equalize contributions from different latitudes.

## Estimation of thermodynamic versus dynamic scaling rate

A physical scaling diagnostic approach[15] is applied to identify the thermodynamic and dynamic contributions in shaping extreme precipitation. Equation (1) expresses the extreme precipitation ($P_e$) with $\omega$ and $q_s$ at constant saturation equivalent potential temperature $\theta^*$. The daily surface pressure, daily vertical wind velocity and air temperature at 20 pressure levels (19 pressure levels for GCMs) conditioned on 99th-percentile precipitation extremes are used to force Eq. (1). Relative humidity does not emerge in these equations, but it can affect precipitation extremes through the dynamic controls.

For each pressure level, $q_s$ is calculated from a modified Tetens formula[54]. The scaling relationship is then used to decompose the change (precipitation extremes conditioned on 12 temperature bins) in thermodynamic and dynamic contributions. For the thermodynamic condition, we use the temporal average vertical velocity $\omega_{avg}$ associated with the extreme precipitation to replace $\omega$ in Eq. (1):

$$P_{the} \sim - \left\{ \omega_{avg} \frac{dq_s}{dp} \Big|_{\theta^*} \right\} \qquad (5)$$

where $P_{the}$ denotes extreme precipitation forcing only by thermodynamic control.

The extreme scaling due to total versus only thermodynamic conditions is then regressed. Finally, the dynamic contribution of extreme precipitation scaling is estimated by subtracting the thermodynamic rates from the full scaling:

$$P_{r-dy}(\delta\omega) = P_r(\delta \ln P_e) - P_{r-the}(\delta \ln P_{the}) \qquad (6)$$

where $P_{r-dy}$ (mainly associated with variations in vertical pressure velocity, $\delta\omega$) and $P_{r-the}$ are dynamic and thermodynamic decomposition, respectively.

To provide further insight into thermodynamic mechanisms, the moist-adiabatic derivative of saturation specific humidity $\frac{dq_s}{dp}\big|_{\theta^*}$ is decomposed as the sum of partial derivative of pressure in saturation specific humidity ($\frac{\partial q_s}{\partial p}$) and the product of the gradient with respect to temperature in saturation specific humidity and the lapse rate ($\frac{\partial q_s}{\partial T} \cdot \frac{dT}{dp}\big|_{\theta^*}$), then thermodynamic extreme precipitation $P_{the}$ can be estimated as:

$$P_{the} \sim - \left\{ \omega_{avg} \frac{dq_s}{dp} \Big|_{\theta^*} \right\} = - \left\{ \omega_{avg} \left( \frac{\partial q_s}{\partial p} + \frac{\partial q_s}{\partial T} \cdot \frac{dT}{dp} \Big|_{\theta^*} \right) \right\} = - \left\{ \omega_{avg} \frac{\partial q_s}{\partial p} \right\} - \left\{ \omega_{avg} \frac{\partial q_s}{\partial T} \cdot \frac{dT}{dp} \Big|_{\theta^*} \right\} \qquad (7)$$

$$pPR = - \left\{ \omega_{avg} \frac{\partial q_s}{\partial p} \right\} \qquad (8)$$

$$pTpLR = - \left\{ \omega_{avg} \frac{\partial q_s}{\partial T} \cdot \frac{dT}{dp} \Big|_{\theta^*} \right\} \qquad (9)$$

To further extract the lapse rate impact ($LR$) from $pTpLR$, we use temporal average lapse rate ($\frac{dT}{dp}\big|_{\theta^*})_{avg}$ in Eq. (9) and obtain the temperature component ($pT$):

$$pT = - \left\{ \omega_{avg} \frac{\partial q_s}{\partial T} \cdot \left( \frac{dT}{dp} \Big|_{\theta^*} \right)_{avg} \right\} \qquad (10)$$

Then $pLR$ can be estimated by using $pTpLR$ minus $pT$:

$$pLR = - \left\{ \omega_{avg} \frac{\partial q_s}{\partial T} \cdot \frac{dT}{dp} \Big|_{\theta^*} \right\} - \left( - \left\{ \omega_{avg} \frac{\partial q_s}{\partial T} \cdot \left( \frac{dT}{dp} \Big|_{\theta^*} \right)_{avg} \right\} \right) \qquad (11)$$

$$P_{the} \sim pPR + pTpLR$$
$$pLR \sim pTpLR - pT \qquad (12)$$
$$P_{the} \sim pPR + pT + pLR$$

Finally, the precipitation-temperature scaling can be computed as the sum of the one dynamic and the three thermodynamic components:

$$P_r(\delta \ln P_e) = P_{r-DY}(\delta\omega) + P_{r-pPR}(\delta \ln pPR) + P_{r-pT}(\delta \ln pT) + P_{r-pLR}(\delta \ln pLR) \qquad (13)$$

## Data availability

The CMIP6 model simulations (pressure levels and single levels) can be downloaded at https://esgf-node.llnl.gov/search/cmip6/. The CMIP5 model simulations (pressure levels and single levels)can be downloaded at https://esgf-node.llnl.gov/search/cmip5/. The ERA5 hourly data on pressure levels can be downloaded from https://cds.climate.copernicus.eu/cdsapp#!/dataset/reanalysis-era5-pressure-levels?tab=overview, and the ERA5 data on single levels are from https://cds.climate.copernicus.eu/cdsapp#!/dataset/reanalysis-era5-single-levels?tab=overview.

## Code availability

The Python (version 3.9) code used for Figs. 1–7 and the MATLAB (version 2021a) code used for data analysis are available at the repository in the Open Science Framework (https://osf.io/wjy7x/).

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

## Acknowledgements

L.G. is supported by the National Natural Science Foundation of China (Grant 52209020). J.Y. is supported by the National Natural Science Foundation of China (Grant No. 52009091; 52242904; 52261145744) and the Fundamental Research Funds for the Central Universities (No. 2042022kf1221). L.J.S. is supported by UKRI (MR/V022008/1) and NERC (NE/S015728/1). This study is also supported by the Research Funds for the State Key Laboratory of Water Resources and Hydropower Engineering Science (2021SWG02) and the National Key Research and Development Program of China (2021YFC3200301). The numerical calculations in this paper have been performed on the supercomputing system in the Supercomputing Centre of Wuhan University.

## Author contributions

L.G. and J.Y. conceived the study, performed the analyses, built the mechanism, and wrote the paper. P.G., H.-M.W., L.S., and S.S. provided critical input and assisted in interpretation of the results. S.G., J.Z., and J.C. contributed to the discussion and improving the paper. L.G., J.Y., P.G., H.-M.W., L.S., S.S., J.C., J.Z., and S.G. reviewed and edited the paper.

## Competing interests

The authors declare no competing interests.
