## [Peer Review File · Nature Communications]

Large anomalies in future extreme precipitation sensitivity driven by atmospheric dynamicsREVIEWER COMMENTS

Reviewer #1 (Remarks to the Author):

Large anomalies in future extreme precipitation sensitivity driven by atmospheric dynamics

by Lei Gu et al.

0. Key Results

First of all, I always associate atmospheric dynamics with regional circulation patterns and less primary with vertical velocity.

The authors studied the sensitivity of extreme precipitation as a function of temperature. Since the underlying physical processes are still not fully understood, global reanalysis data and climate model simulations were diagnostically broken down into a dynamic and thermodynamic parts. They found out that the thermodynamic component under certain dynamic conditions (updrafts) does not always lead to an increase in extreme precipitation. Ocean areas tend to show an increase and land areas a decrease. A large uncertainties in future projections of extreme precipitation can be attributed to the dynamical component. The review of the manuscript yielded only a few comments and suggestions for improvement. It's well written and reviewer votes to minor revision.

There are no fundamentals to complain about. This study shows once again the complexity of dealing with extreme precipitation and its possible influencing factors considering physical quantities across atmospheric layers. However, the horizontal and synoptical patterns remain underlit. That's pity. The main driver for the large uncertainty is due to the diversity of synoptical patterns and their respective vertical characteristics. More in ERA5 than in CMIP6. In order to better understand processes, a further decomposition into synoptical patterns is necessary.

1. Abstract

- No comment

2. Introduction

- L39: Main Text -> Introduction

- No further comments

3. Results:

- Q1: The actual extreme precipitation is estimated by the 99th percentile for a time period of 30 years and daily values. Why, only days above 0.1mm/d are considered? This threshold can be biased between ERA5 and CMIP6! For the binning-scaling method you also calculate the 99th percentile for each bin. What exactly is shown in Figure 1 (only wet days)?

- Q2: Do I understand correctly that if you only look at wet days (>0.1mm/d) a rough synoptic distinction is made. Wet days are usually determined by cyclonic weather condition.

- L59: Variations in large-scale atmospheric circulation and local weather patterns are not directly considered, right? Is there any linkage between the dynamical component (vertical velocity) and synoptical phenomena? What synoptical patterns represent the 99th percentile of extreme rainfall?

- L100: vertical temperature -> temperature

4. Discussion:

- Q3: It's a bit exhausting when you have jump quite often to the supplement materials in the discussion part. I don't know, how to change this.

- L386: pressure air temperature?

- Q4: Is there any fundamental changes in the scaling regimes between CMIP6 and CMIP5?

- Q5: Can you explain again the main effect on EPS in hook regimes, when the local temperature increase is higher/lower than the increase of Tpp?

- Q6: Is there a physical explanation, why strongly positive EPS anomalies emerge in oceans and negative anomalies occur in land areas?

5. Figures:

- Fig.1: L116 (diagnostic-based minus actual 99th-percentile precipitation) can be removed?
- Fig.6a,d,g: why the y-label "99th percentile precipitation" ranges from negative to positive?
- Fig.6a-i: All subfigures should have the same range in x and y dimension.
- Fig.2 & Fig.4: I suggest a nonlinear intervall $[-16,-8,-4,-2,-1,1,2,4,8,16]$
- Supplementary Fig.7: Tpp-Tas for ERA5 is missing

6. Supplements:

- In the supplementary document a brief abstract of the main content of the additional material is missing and would be helpful beyond the capture.

Reviewer #2 (Remarks to the Author):

The manuscript entitled "*Large anomalies in future extreme precipitation sensitivity driven by atmospheric dynamics*" by Gu et al. focuses on the Extreme Precipitation Sensitivity (EPS) to temperature, which is a very interesting topic, gathering scientific attention. In this work, they build on the framework of O’Gorman & Schneider (2009), which separates the thermodynamic and dynamic components of precipitation extremes increase, and they extend it by separating the thermodynamic component into three separate sub-components, which to my knowledge, is a novelty.

Overall, I find this work very interesting and the authors use both reanalysis and CMIP6 data, to demonstrate the robustness of the approach. The manuscript is clearly written and nicely presented, while all the findings are properly reported. All the necessary information to interpret the results is given in the manuscript and the accompanying Supplementary Material.

Considering the above, I believe this manuscript can be a valuable contribution to the scientific discussion on the issue, and I can recommend this for publication upon some slight improvement.

Below the authors can find some comments that could possibly help improve the methodology of estimating the scaling of precipitation extremes with surface air temperature, and some more general remarks and ideas that could possibly help improve the manuscript and increase its impact.

Comments:

1. **On the binning method:** The authors use the binning method, even though it has been highly criticized and it has been demonstrated that quantile regression is a much more suited tool for this work (Wasko & Sharma 2014). Quantile regression does not necessarily have to assume the existence of a monotonic relation but can also be used in a scheme where the existence of a hook-like structure is investigated. We did this, for example, in our previous work in Moustakis et al. (2020). I think switching to a quantile-regression based approach would make your results more robust.
2. **On the detection of the hook-like structure:** With the methodology used by the authors, the existence of the hook-like structure can be strongly biased by the choice of the temperature bins, to my understanding. If it “happens” that the higher rainfall events belong to the higher bin (which the user has arbitrarily defined), then you end up with a monotonically increasing relationship, whereas, if they don’t, you end up with a hook-like structure. How do the authors make sure that the detection of a hook-like structure is robust enough and not biased by the methodology used? This is an important issue, since the hook-like structure is a major focus of this work. To address this, I would again advise the authors to avoid using the binning method, and result to quantile-regression based techniques. As an illustrative example, in our work (Moustakis 2020) we used piece-wise quantile regression and the Akaike information criterion to determine whether a hook-like structure is better at capturing the scaling behavior, compared to a monotonic one.
3. **On the issue of convection:** Coarse resolution models (such as the model used to produce ERA5, and GCMs) are known to not explicitly resolve atmospheric convection, thus underestimating convective precipitation extremes. But also, they underestimate rainfall extremes in general. What we showed in our recent work (Moustakis 2020) is that convective events (which usually occupy higher temperature “bins”) can be underestimated more strongly compared to non-convective ones (which mostly occupy lower temperature “bins”), and this yielded a hook-like structure, which was not necessarily present in rain gauge data. I think this issue can be

discussed, given that the hook-like structure is a big emphasis of this work. Given that the data are readily available and used by the authors, maybe the authors could characterize the thermodynamic environments associated with each event (e.g., Convective Available Potential Energy, Convective Inhibition etc.), without too much effort, and try to associate the existence of convective events to the hook-like structure, and see if this is indeed the case over regions where strong non-convective and convective events co-exist.

4. **On the discussion:** To my understanding the decomposition of the thermodynamic term in sub-components is a major novelty of this work that should be highlighted more in the discussion, despite the fact that the lapse rate and pressure terms turned out to play a rather small role. Maybe the authors could write a paragraph on whether they expected this to be the case, and, whether, for example, they would expect these roles to be more significant in a convection-permitting setup.
5. **On the choice of models:** It is not entirely clear why the authors end up with these specific models, and only 1 (and 2 in one case) ensemble member for each model. Was this an issue of data unavailability? If not, then I see no particular reason why the authors should not extend their analysis to include more data.
6. **On ERA5:** To my knowledge rainfall is a model output in ERA5. If this is still the case in version 5, then the authors should clarify this in the Methods section, so that we make sure that the readers know this is not an “observational” dataset.
7. **On the temporal resolution:** Maybe the authors could consider performing the same analysis but at the hourly resolution (at least for ERA5 where hourly data are available). It would be really interesting to see whether the contributions of the different components change at this resolution, and maybe this would help gain more insights on the results.

Minor comments:

- **Title:** Precipitation sensitivity refers to temperature, but it is not entirely clear in the title. Maybe consider changing that.
- Line 297: “The dynamic term plays”

Sincerely,

Dr. Yiannis Moustakis

Chair for physical geography and land use systems

Department of Geography

Ludwig-Maximilians Universität Munich

References:

Moustakis, Y., Onof, C.J. & Paschalis, A. Atmospheric convection, dynamics and topography shape the scaling pattern of hourly rainfall extremes with temperature globally. *Commun Earth Environ* 1, 11 (2020). <https://doi.org/10.1038/s43247-020-0003-0>

O’Gorman, Paul A., and Tapio Schneider. "The physical basis for increases in precipitation extremes in simulations of 21st-century climate change." *Proceedings of the National Academy of Sciences* 106, no. 35 (2009): 14773-14777.

Wasko, C., and Sharma, A. (2014), Quantile regression for investigating scaling of extreme precipitation with temperature, *Water Resour. Res.*, 50, 3608– 3614, doi:10.1002/2013WR015194.

Reply to Reviewers' comments

Legend

Reviewers' comments

Authors' responses

Direct quotes from the revised manuscript

Reviewer #1 (Remarks to the Author):

Large anomalies in future extreme precipitation sensitivity driven by atmospheric dynamics by Lei Gu et al.

0. Key Results

First of all, I always associate atmospheric dynamics with regional circulation patterns and less primary with vertical velocity.

The authors studied the sensitivity of extreme precipitation as a function of temperature. Since the underlying physical processes are still not fully understood, global reanalysis data and climate model simulations were diagnostically broken down into a dynamic and thermodynamic parts. They found out that the thermodynamic component under certain dynamic conditions (updrafts) does not always lead to an increase in extreme precipitation. Ocean areas tend to show an increase and land areas a decrease. A large uncertainties in future projections of extreme precipitation can be attributed to the dynamical component. The review of the manuscript yielded only a few comments and suggestions for improvement. It's well written and reviewer votes to minor revision. There are no fundamentals to complain about. This study shows once again the complexity of dealing with extreme precipitation and its possible influencing factors considering physical quantities across atmospheric layers. However, the horizontal and synoptical patterns remain underlit. That's pity. The main driver for the large uncertainty is due to the diversity of synoptical patterns and their respective vertical characteristics. More in ERA5 than in CMIP6. In order to better understand processes, a further decomposition into synoptical patterns is necessary.

Reply: We appreciate the reviewer's positive evaluation and the insightful comments on the improvement of the manuscript. All your concerns have been addressed; we sincerely hope the revised manuscript satisfies these concerns to be published this time.

We fully understand your attention on synoptic patterns. As our study mainly focuses on decomposing the thermodynamic terms, our previous version did not pay much attention to synoptic mechanisms. To systematically address your concerns, we have elaborated on the impacts of convection and large-scale weather patterns on extreme precipitation sensitivity to local temperature. Details can be found in the revised

Discussion Section as follows:

The mechanisms behind extreme precipitation scaling are quite complex in some regions. In tropical regions where EPS is governed by the dynamic term, extreme precipitation is typically associated with storms and cyclones. Other synoptic patterns, including moisture transport from low level jets and upper-level atmospheric rivers, also play a role in modulating EPS⁴⁰. In mid-latitudes such as the Southeast and Midwestern US, Southeast China, and Southern Australia, deep convection dominates extreme precipitation, as indicated by very large convective available potential energy (CAPE) and convective inhibition (CIN) anomalies (Supplementary Figure 12a-b). This convection is accompanied by high total column water vapor and strong moisture convergence during extreme precipitation (Supplementary Figure 12c-d). Interestingly, these regions all firmly exhibit a hook-like scaling, using both the binning scaling and quantile regression approaches, at both hourly and daily temporal scales. The physics behind this hook-like structure is multifaceted. Specifically, we find it is determined by the reduced sensitivity of vertical velocity at higher temperatures. Another explanation is the substantial underestimation of convective events at higher temperatures^{35,36}. The CAPE corresponding to extreme precipitation is increasing almost monotonically with rising temperature over these regions (Supplementary Figure 13). However, reduced moisture availability and decreased relative humidity in warmer environments may weaken moisture convergence (Supplementary Figure 13) and eventually lead to decreasing precipitation intensity at high temperatures³⁸. In higher latitudes with a monotonically increasing scaling (e.g., in Europe), extreme precipitation is more dependent on low pressure systems and atmospheric rivers than convection and is impacted more by the thermodynamic terms than the dynamic contribution³⁰. Current generation of climate models are accompanied by subgrid-scale uncertainties due to their coarse resolution, and future studies could combine climate models with machine-learning techniques to further explore the decomposition of EPS at sub-grid cloud-resolving scales⁴¹.

Supplementary Figure 12. Anomaly of the thermodynamic and dynamic variables corresponding to extreme precipitation during 1985-2014 from ERA5 dataset. a, Convective available potential energy (CAPE) anomalies corresponding to extreme precipitation. Here anomalies denote CAPE values during extreme precipitation minus temporal mean values during the reference period. **b,** Convective inhibition (CIN) anomalies corresponding to extreme precipitation. **c,** Total column water vapor (TCWV) anomalies corresponding to extreme precipitation. **d,** Vertically integrated moisture divergence (VIMD) anomalies corresponding to extreme precipitation. Negative values denote moisture convergence.

Supplementary Figure 13. The thermodynamic and dynamic variables scaling with rising temperature during 1985-2014 from ERA5 dataset. a-d, the scaling rate of CAPE, CIN, TCWV and VIMD with rising temperature. **e-h**, the T_{pp} of CAPE, CIN, TCWV and VIMD. **i-l**, the zonal median scaling rate of CAPE, CIN, TCWV and VIMD. Here, all these variables are corresponding to 99th percentile daily precipitation.

30. Chan, S. C., Kendon, E. J., Roberts, N. M., Fowler, H. J. & Blenkinsop, S. Downturn in scaling of UK extreme rainfall with temperature for future hottest days. *Nat. Geosci.* **9**, 24-28, (2016).

35. Moustakis, Y., Onof, C. J. & Paschalis, A. Atmospheric convection, dynamics and topography shape the scaling pattern of hourly rainfall extremes with temperature globally. *Communications Earth & Environment* **1**, (2020).

36. Moustakis, Y., Papalexiou, S. M., Onof, C. J. & Paschalis, A. Seasonality, Intensity, and Duration of Rainfall Extremes Change in a Warmer Climate. *Earth's Future* **9**, (2021).

38. Meredith, E. P., Ulbrich, U. & Rust, H. W. The Diurnal Nature of Future Extreme Precipitation Intensification. *Geophys. Res. Lett.* **46**, 7680-7689, (2019).

40. Fowler, H. J. *et al.* Anthropogenic intensification of short-duration rainfall extremes. *Nat. Rev. Earth. Environ.* **2**, 107-122, (2021).

41. Rasp, S., Pritchard, M. S. & Gentine, P. Deep learning to represent subgrid processes in climate models. *Proc. Natl Acad. Sci.* **115**, 9684-9689, (2018).

1. Abstract
 No comment
 2. Introduction

L39: Main Text -> Introduction

No further comments

Reply: Thanks for these positive comments.

3. Results:

Q1: The actual extreme precipitation is estimated by the 99th percentile for a time period of 30 years and daily values. Why, only days above 0.1mm/d are considered? This threshold can be biased between ERA5 and CMIP6! For the binning-scaling method you also calculate the 99th percentile for each bin. What exactly is shown in Figure 1 (only wet days)?

Reply: Following many previous studies (e.g., Kendon et al., 2014; Ban et al., 2015; Yin et al., 2018), we choose this threshold to consider only wet-day 99th percentile extremes (as shown in Fig. 1). Moreover, the high similarity of 99th percentile daily extreme precipitation between the ERA5 and CMIP6 outputs (Fig. 1a,c) indicates that the threshold contribute to small biases.

Ban, N., Schmidli, J., & Schär, C. (2015). Heavy precipitation in a changing climate: Does short-term summer precipitation increase faster?. Geophysical Research Letters, 42(4), 1165-1172.
Kendon, E. J., N. M. Roberts, H. J. Fowler, M. J. Roberts, S. C. Chan, and C. A. Senior (2014), Heavier summer downpours with climate change revealed by weather forecast resolution model, Nat. Clim. Change, 4, 570– 576, doi:10.1038/nclimate2258.
Yin, J., Gentine, P., Zhou, S., Sullivan, S. C., Wang, R., Zhang, Y., & Guo, S. (2018). Large increase in global storm runoff extremes driven by climate and anthropogenic changes. Nature communications, 9(1), 4389.

- Q2: Do I understand correctly that if you only look at wet days (>0.1mm/d) a rough synoptic distinction is made. Wet days are usually determined by cyclonic weather condition.

Reply: Yes, we consistently focusing on wet days across the manuscript. We have briefly explained this in the revised **Methods** as follows:

Within each temperature bin (more than 150 precipitation events in each bin), the daily precipitation series is used to estimate the 99th percentile, and the three nearest events to this 99th percentile are averaged to define the daily extreme. In estimating the 99th percentile, we only employ wet days (precipitation >0.1mm/d) to focus on the intensity rather than the frequency of precipitation, as only the intensity can scale with saturation vapor pressure (Ban et al., 2015).

- L59: Variations in large-scale atmospheric circulation and local weather patterns are not directly considered, right? Is there any linkage between the dynamical component

(vertical velocity) and synoptical phenomena? What synoptical patterns represent the 99th percentile of extreme rainfall?

Reply: Correct; we did not directly analyze the large-scale atmospheric circulation and local weather patterns in this study. To fully address your concerns, we have analyzed the CAPE, CIN, TCWV and VIMD conditions corresponding to extreme precipitation and we discuss the synoptic patterns associated with the 99th percentile of extreme precipitation can be found in the supplementary Figures 12-13 and response to **Q1**.

- L100: vertical temperature -> temperature

Reply: Done.

4. Discussion:

- Q3: It's a bit exhausting when you have jump quite often to the supplement materials in the discussion part. I don't know, how to change this.

Reply: Thanks for this comment. We have added a brief abstract to introduce the supplement figures in the supplement materials:

Summary about supplementary information

This supplementary file includes 17 figures and 2 tables:

Supplementary Fig. 1 evaluates the robustness of the physical diagnostic approach in estimating EPS during the historical and future periods.

Supplementary Fig. 2 verifies the robustness of the binning scaling approach by using the quantile regression and the Akaike information criterion.

Supplementary Figs. 3-5 serve for the “Decomposition of thermodynamic and dynamic contributions” section. We demonstrate the thermodynamic versus dynamic scaling (Supplementary Fig. 2) and further show the decomposed three thermodynamic (i.e., pPR , pT and pLR) and one dynamic (i.e., DY) scaling rates (Supplementary Fig. 3) as well as associated T_{pp} (Supplementary Fig. 4) during the historical period.

Supplementary Figs. 6-11 serve for the “Shifting thermodynamic and dynamic

controls under climate change” section. We firstly analyze the T_{pp} changes (Supplementary Fig. 5) and local mean temperature (T_m) changes (Supplementary Fig. 6) between the 1985-2014 reference and 2071-2100 future periods. We then compare differences between T_{pp} and T_m during reference (Supplementary Fig. 7) and future (Supplementary Fig. 8) periods, respectively, to fully investigate local warming impacts on the shifting EPS. Finally, we decompose the EPS and T_{pp} changes between 1985-2014 reference and 2071-2100 future periods into three thermodynamic and one dynamic components (Supplementary Figs. 9-10) to show how each term contributes to the total changes.

Supplementary Figs. 12-13 explore the synoptic patterns behind extreme precipitation. We use four indices (including CAPE, CIN, TCWV and VIMD) to analyze the physics behind extreme precipitation, including estimate their anomalies during extreme precipitation (Supplementary Fig. 12) and their scaling with rising temperature (Supplementary Fig. 13).

Supplementary Fig. 14 explore the differences between CMIP5 and CMIP6 results. We use the CMIP5 outputs to perform the study and present the results in Supplementary Fig. 14.

Supplementary Figs. 15-17 serve for the “Discussion” section. We investigate the uncertainty of T_{pp} (Supplementary Fig. 11) and EPS (Supplementary Fig. 12) changes between 1985-2014 reference and 2071-2100 future periods and find the dynamic term plays a dominant role. We also analyze future changes in precipitation and runoff (Supplementary Fig. 13) to show the shifting EPS impacts on the water cycle.

Supplementary Tables. 1-2 demonstrate the information of CMIP6 (Supplementary Table. 1) and CMIP5 (Supplementary Table. 2) ensembles used in the study.

- L396: *pressure air temperature?*

Reply: We have revised it as “air temperature”.

- Q4: Is there any fundamental changes in the scaling regimes between CMIP6 and CMIP5?

Reply: CMIP5 only provides a rough estimation of daily vertical profiles varying from 8 to 15 pressure levels, while CMIP6 enables a much more detailed characterization to vertical variables (e.g., air temperature and vertical velocity) which are available at 19 pressure levels. To explore the differences between CMIP5 and CMIP6, we have downloaded six CMIP5 model outputs (to keep the same model size as in the current study) to reproduce the scaling regimes and compared them with the CMIP6 results. A brief discussion is included as follows:

We also decompose the EPS and EPS anomalies using CMIP5 outputs (see Supplementary Table S2) for comparison. In the mid-to-high latitudes, extreme precipitation scaling in the CMIP5 models is similar with the scaling in the CMIP6 models during the historical period (Supplementary Fig. 14a-c). The main difference in the historical EPS between CMIP5 and CMIP6 lies in the ITCZ, where the strong negative scaling seen in both ERA5 and CMIP6 disappear. The lack of negative scaling in the tropical region in CMIP5 may be attributed to different parameterization schemes in the models. When comparing future EPS anomalies, the varying regimes across different latitudes found in CMIP6 hold in CMIP5 (Supplementary Fig. 14d-e), although the associated changes in T_{pp} are slightly smaller in CMIP5 than in CMIP6. Overall, most results in CMIP5, including the EPS, EPS anomalies and their decomposition, mirror those in CMIP6 (Supplementary Fig. 14), adding further credence to our conclusions.

Supplementary Table 2. Basic information about the available CMIP5 model outputs

NO	Institution	Model name	Ensemble	Resolution
				Lon.×Lat.×Plev.
1	Canadian Centre for Climate Modelling and Analysis	CanESM2	r1i1p1f1	128×64×8
2	Centro Euro-Mediterraneo sui Cambiamenti Climatici	CMCC-CESM	r1i1p1f1	96×48×11
3	NOAA Geophysical Fluid Dynamics Laboratory	GFDL-CM3	r1i1p1f1	144×90×8
4	Japan agency for marine-earth science and technology	MIROC5	r1i1p1f1	180×90×8
5	Institut Pierre Simon Laplace	IPSL-CM5A-MR	r1i1p1f1	180×90×8

Supplementary Figure 14. EPS, T_{pp} and dominant factor during the reference 1985-2014 period (a-c) and EPS anomalies, T_{pp} change and associated dominant factor during the future 2071-2100 future period relative reference period (d-f). **g**, Zonal EPS and the DY , pPR , pT and pLR contributions to zonal EPS during the reference period. **h**, Zonal EPS anomalies and the associated DY , pPR , pT and pLR contributions to zonal EPS anomalies during the future 2071-2100 future period relative reference period.

- *Q5: Can you explain again the main effect on EPS in hook regimes, when the local temperature increase is higher/lower than the increase of T_{pp} ?*

Reply: Sorry to the vague description. We have re-clarified this section in the revised manuscript as follows:

The changes in T_{pp} are however slower than local warming rates (defined as future minus reference local mean temperature, T_m), with global spatially averaged increasing values of 3.5°C and 4.6°C, respectively (Supplementary Figs. 6-7). If T_m does indeed exceed T_{pp} , the EPS regime may shift to a descending scaling, potentially mitigating future extreme precipitation intensification. However, T_m remains substantially lower

than T_{pp} in both reference and future climates across most regions of the globe (Supplementary Figs. 8-9), even though the difference between T_m and T_{pp} might shrink due to faster increases in T_m under future climate warming. Most regions still exhibit positive EPS scaling as a result, and future precipitation is likely to intensify by the end of 21st century.

- Q6: *Is there a physical explanation, why strongly positive EPS anomalies emerge in oceans and negative anomalies occur in land areas?*

Reply: It's really a good point to further clarified. We have elaborated the physical mechanisms in the revised section as follows:

Large positive zonal average EPS anomalies mainly occur between $\sim 30^\circ\text{S}$ and $\sim 30^\circ\text{N}$ (Fig. 5a), and are mainly contributed by the positive EPS anomalies over the oceans, while the land areas consistently demonstrate negative EPS anomalies in the latitude bands (Fig. 5f,k). These large land-ocean discrepancies of EPS anomalies can mainly be attributed to moisture limitation. In the context of a future warming climate, extreme precipitation is projected to increase sharply over the oceans with rising temperature, given sufficient moisture supply. Over land areas, although saturation vapor pressure still strongly increases with warming, enhanced vapor pressure deficit results from moisture limitation can constrain extreme precipitation intensification.

From the decomposition, the dynamic component explains the large anomalies at low latitudes (Fig. 6b,g,l). Indeed, it controls EPS anomalies across $\sim 80.9\%$ of the globe (Fig. 5g,h) and is larger than any of the thermodynamic components.

5. *Figures:*

- Fig.1: L116 (diagnostic-based minus actual 99th-percentile precipitation) can be removed?

Reply: We have removed the content in the bracket to simplify the figure caption.

- Fig.6a,d,g: why the y-label "99th percentile precipitation" ranges from negative to positive?

Reply: Since the dynamic term does not always contribute positively to precipitation intensification, there are negative values in the figure, indicating the dynamic term could reduce extreme precipitation in hot environments.

- Fig.6a-i: All subfigures should have the same range in x and y dimension.

Reply: Thanks for this suggestion. The x and y axes have been kept consistent across horizontal subfigures:

Fig. 7 | P_e - T scaling and associated component contribution during reference and future periods based on CMIP6 for the three regimes. a,d,g Dynamic versus thermodynamic controls on P_e - T scaling. b,e,h The vertical velocity (ω) over 12 precipitation-temperature bins. The blue

(grey) range is estimated by the minimum and maximum vertical velocity in 12 bins for the historical (His) and future (Fut) periods, whereas the blue dashed (solid black) line indicates the vertical velocity corresponding to the T_{pp} . **c,f,i** The contribution from one dynamic (DY) and three thermodynamic (pPR , pT , pLR) components to P_e - T scaling during the 1985-2014 (His) and 2071-2100 (Fut) period, respectively.

- Fig.2 & Fig.4: I suggest a nonlinear interval $\backslash[-16,-8,-4,-2,-1,1,2,4,8,16]$

Reply: Thanks and done, please see the revised Figs. 2 & 5 as follows:

Fig. 2 | EPS based on ERA5 and CMIP6 during the reference 1985-2014 period. a-b 99th-percentile precipitation-temperature scaling rate based on ERA5 reanalysis before T_{pp} ($< T_{pp}$, exhibiting three EPS regimes) and after the T_{pp} ($>T_{pp}$, i.e. only the decreasing branch in the hook-like scaling). **c-d** Results based on CMIP6 average multi-model ensemble experiments. Monotonic scaling types (monotonically increasing and decreasing regimes) in **(b,d)** are masked in grey.

Fig. 5 | Contribution of thermodynamic versus dynamic components to EPS anomalies and T_{pp} changes between the 1985-2014 reference and 2071-2100 future periods. a,d EPS anomalies (i.e., relative to the reference period) before ($<T_{pp}$) and after T_{pp} ($>T_{pp}$), respectively. **b,e** Thermodynamic contribution to EPS anomalies before and after T_{pp} . **c,f** Dynamic contribution to EPS anomalies before and after T_{pp} . **g,h** The dominant factor (showing the greatest contribution among DY , pPR , pT and pLR components) contributing to EPS anomalies before and after T_{pp} . The monotonically increasing and decreasing regimes (without an additional decreasing branch) are masked in grey in (d-f,h). **i** T_{pp} changes projected by CMIP6 multi-model ensemble mean. T_{pp} changes are only presented in the hook-like regime spanning both reference and future periods. Otherwise, in locations corresponding to the monotonically increasing and decreasing regimes, the changing behaviors are masked in grey.

- *Supplementary Fig.7: $T_{pp}-T_m$ for ERA5 is missing*

Reply: In Supplementary Fig. 7, we originally focused on $T_{pp}-T_m$ from climate modelling, in comparison to results for the 2071-2100 future period (Supplementary Fig. 8) and thus neglected $T_{pp}-T_m$ from ERA5. However, considering the reviewer's suggestion, we have added $T_{pp}-T_m$ for ERA5 in the revised Supplementary Fig. 8 as follows:

Supplementary Figure 8. Differences between T_{pp} and T_m during 1985-2014 reference period within CMIP6 and ERA5 datasets. (a) Differences (T_{pp} minus T_m) simulated by CanESM5 model. **(b-g)** The same as (a), but by HadGEM3-GC31-LL, INM-CM4-8, INM-CM5-0, MIROC6, UKESM1-0-LL (*r1*) and UKESM1-0-LL (*r14*), respectively. Basic model information is shown in Table S1. **(h)** The same as (a), but for multi-model ensemble mean results (MME). **(i)** The same as (a), but for ERA5 dataset. The results are only for regions showing a hook structure spanning both reference and future periods. Otherwise, in areas characterized by the monotonically increasing and decreasing regimes, the changing behaviors are masked by grey.

6. *Supplements:- In the supplementary document a brief abstract of the main content of the additional material is missing and would be helpful beyond the capture.*

Reply: Thanks for this useful suggestion. We have added a brief abstract before the supplementary figures, please see response to **Q3**.

Reviewer #2 (Remarks to the Author):

Dear Editor,

The manuscript entitled “Large anomalies in future extreme precipitation sensitivity driven by atmospheric dynamics” by Gu et al. focuses on the Extreme Precipitation Sensitivity (EPS) to temperature, which is a very interesting topic, gathering scientific attention. In this work, they build on the framework of O’Gorman & Schneider (2009), which separates the thermodynamic and dynamic components of precipitation extremes increase, and they extend it by separating the thermodynamic component into three separate sub-components, which to my knowledge, is a novelty. Overall, I find this work very interesting and the authors use both reanalysis and CMIP6 data, to demonstrate the robustness of the approach. The manuscript is clearly written and nicely presented, while all the findings are properly reported. All the necessary information to interpret the results is given in the manuscript and the accompanying Supplementary Material. Considering the above, I believe this manuscript can be a valuable contribution to the scientific discussion on the issue, and I can recommend this for publication upon some slight improvement. Below the authors can find some comments that could possibly help improve the methodology of estimating the scaling of precipitation extremes with surface air temperature, and some more general remarks and ideas that could possibly help improve the manuscript and increase its impact.

Reply: Dear Dr. Moustakis, we appreciate your insightful suggestions. All your concerns have been addressed in the revised manuscript; we sincerely hope the revised manuscript is satisfactory for publication this time.

Comments:

1. On the binning method: The authors use the binning method, even though it has been highly criticized and it has been demonstrated that quantile regression is a much more suited tool for this work (Wasko & Sharma 2014). Quantile regression does not necessarily have to assume the existence of a monotonic relation but can also be used in a scheme where the existence of a hook-like structure is investigated. We did this, for example, in our previous work in Moustakis et al. (2020). I think switching to a quantile-regression based approach would make your results more robust.

Reply: As the reviewer suggested, we have used the quantile regression method to verify the robustness of our results based on the daily ERA5 dataset (Supplementary Fig. 2), and the results are added in the revised Decomposition of thermodynamic and dynamic contributions Section as follows:

We further explore the robustness of the binning scaling method in detecting EPS by using a quantile-regression based technique instead^{35,36}. The results are very similar: most mid-latitudes show a hook-like structure, low latitudinal regions present a negative scaling and high latitudes are dominated by monotonically increasing scaling (Supplementary Fig. 2).

Supplementary Figure 2. EPS and T_{pp} based on the quantile regression and the Akaike information criterion during the reference 1985-2014 period using ERA5 dataset.

35. Moustakis, Y., Onof, C. J. & Paschalis, A. Atmospheric convection, dynamics and topography shape the scaling pattern of hourly rainfall extremes with temperature globally. *Communications Earth & Environment* **1**, (2020).
36. Moustakis, Y., Papalexiou, S. M., Onof, C. J. & Paschalis, A. Seasonality, Intensity, and Duration of Rainfall Extremes Change in a Warmer Climate. *Earth's Future* **9**, (2021).

2. *On the detection of the hook-like structure: With the methodology used by the authors, the existence of the hook-like structure can be strongly biased by the choice of the temperature bins, to my understanding. If it “happens” that the higher rainfall events belong to the higher bin (which the user has arbitrarily defined), then you end up with a monotonically increasing relationship, whereas, if they don’t, you end up with a hook-like structure. How do the authors make sure that the detection of a hook-like structure is robust enough and not biased by the methodology used? This is an important issue, since the hook-like structure is a major focus of this work. To address this, I would again advise the authors to avoid using the binning method, and result to quantile-regression based techniques. As an illustrative example, in our work (Moustakis 2020) we used piece-wise quantile regression and the Akaike information criterion to determine whether a hook-like structure is better at capturing the scaling behavior, compared to a monotonic one.*

Reply: We have used the quantile regression and the Akaike information criterion mentioned in Mouskatis et al. (2020) to detect the scaling behavior of extreme precipitation and find similar results to the binning scaling approach. There are actually more regions (such as northern North America, Europe and Russia) showing monotonically increasing scaling when using the quantile regression method. However, most mid-latitude regions still retain a hook-like scaling, which to some extent confirm the robustness of our binning scaling approach. Details can be found in the response to Q2.

3. *On the issue of convection: Coarse resolution models (such as the model used to produce ERA5, and GCMs) are known to not explicitly resolve atmospheric convection, thus underestimating convective precipitation extremes. But also, they underestimate*

rainfall extremes in general. What we showed in our recent work (Moustakis 2020) is that convective events (which usually occupy higher temperature “bins”) can be underestimated more strongly compared to non-convective ones (which mostly occupy lower temperature “bins”), and this yielded a hook-like structure, which was not necessarily present in rain gauge data. I think this issue can be discussed, given that the hook-like structure is a big emphasis of this work. Given that the data are readily available and used by the authors, maybe the authors could characterize the thermodynamic environments associated with each event (e.g., Convective Available Potential Energy, Convective Inhibition etc.), without too much effort, and try to associate the existence of convective events to the hook-like structure, and see if this is indeed the case over regions where strong non-convective and convective events co-exist.

Reply: Thanks for this constructive suggestion. We have fully investigated the CAPE, CIN, TCWV and VIMD corresponding to precipitation extremes across different temperature bins. We have provided the following discussion in the revised manuscript as follows:

The mechanisms behind extreme precipitation scaling are quite complex in some regions. In tropical regions where EPS is governed by the dynamic term, extreme precipitation is typically associated with storms and cyclones. Other synoptic patterns, including moisture transport from low level jets and upper-level atmospheric rivers, also play a role in modulating EPS⁴⁰. In mid-latitudes such as the Southeast and Midwestern US, Southeast China, and Southern Australia, deep convection dominates extreme precipitation, as indicated by very large convective available potential energy (CAPE) and convective inhibition (CIN) anomalies (Supplementary Figure 12a-b). This convection is accompanied by high total column water vapor and strong moisture convergence during extreme precipitation (Supplementary Figure 12c-d). Interestingly, these regions all firmly exhibit a hook-like scaling, using both the binning scaling and quantile regression approaches, at both hourly and daily temporal scales. The physics behind this hook-like structure is multifaceted. Specifically, we find it is determined by the reduced sensitivity of vertical velocity at higher temperatures. Another explanation is the substantial underestimation of convective events at higher temperatures^{35,36}. The CAPE corresponding to extreme precipitation is increasing almost monotonically with rising temperature over these regions (Supplementary Figure 13). However, reduced moisture availability and decreased relative humidity in warmer environments may

weaken moisture convergence (Supplementary Figure 13) and eventually lead to decreasing precipitation intensity at high temperatures³⁸. In higher latitudes with a monotonically increasing scaling (e.g., in Europe), extreme precipitation is more dependent on low pressure systems and atmospheric rivers than convection and is impacted more by the thermodynamic terms than the dynamic contribution³⁰. Current generation of climate models are accompanied by subgrid-scale uncertainties due to their coarse resolution, and future studies could combine climate models with machine-learning techniques to further explore the decomposition of EPS at sub-grid cloud-resolving scales⁴¹.

Supplementary Figure 12. Anomalies of the thermodynamic and dynamic variables corresponding to extreme precipitation during 1985-2014 from ERA5 dataset. a, Convective available potential energy (CAPE) anomalies corresponding to extreme precipitation. Here anomalies denote CAPE values during extreme precipitation minus temporal mean values during the reference period. **b,** Convective inhibition (CIN) anomalies corresponding to extreme precipitation. **c,** Total column water vapor (TCWV) anomalies corresponding to extreme precipitation. **d,** Vertically integrated moisture divergence (VIMD) anomalies corresponding to extreme precipitation. Negative values denote moisture convergence.

Supplementary Figure 13. The thermodynamic and dynamic variables scaling with rising temperature during 1985-2014 from ERA5 dataset. a-d, the scaling rate of CAPE, CIN, TCWV and VIMD with rising temperature. **e-h**, the T_{pp} of CAPE, CIN, TCWV and VIMD. **i-l**, the zonal median scaling rate of CAPE, CIN, TCWV and VIMD. Here, all these variables are corresponding to 99th percentile daily precipitation.

30. Chan, S. C., Kendon, E. J., Roberts, N. M., Fowler, H. J. & Blenkinsop, S. Downturn in scaling of UK extreme rainfall with temperature for future hottest days. *Nat. Geosci.* **9**, 24-28, (2016).
35. Moustakis, Y., Onof, C. J. & Paschalis, A. Atmospheric convection, dynamics and topography shape the scaling pattern of hourly rainfall extremes with temperature globally. *Communications Earth & Environment* **1**, (2020).
36. Moustakis, Y., Papalexiou, S. M., Onof, C. J. & Paschalis, A. Seasonality, Intensity, and Duration of Rainfall Extremes Change in a Warmer Climate. *Earth's Future* **9**, (2021).
38. Meredith, E. P., Ulbrich, U. & Rust, H. W. The Diurnal Nature of Future Extreme Precipitation Intensification. *Geophys. Res. Lett.* **46**, 7680-7689, (2019).
40. Fowler, H. J. *et al.* Anthropogenic intensification of short-duration rainfall extremes. *Nat. Rev. Earth. Environ.* **2**, 107-122, (2021).
41. Rasp, S., Pritchard, M. S. & Gentine, P. Deep learning to represent subgrid processes in climate models. *Proc. Natl Acad. Sci.* **115**, 9684-9689, (2018).

4. On the discussion: To my understanding the decomposition of the thermodynamic term in subcomponents is a major novelty of this work that should be highlighted more in the discussion, despite the fact that the lapse rate and pressure terms turned out to

play a rather small role. Maybe the authors could write a paragraph on whether they expected this to be the case, and, whether, for example, they would expect these roles to be more significant in a convection permitting setup.

Reply: Thanks for this comment. We have expanded a paragraph in the revised Discussion to discuss the thermodynamic decomposition in greater depth, alongside convection parameterization, as follows:

As the internal terms of atmospheric thermodynamics demonstrate divergent contributions, we present a novel decomposition of EPS into one dynamic and three thermodynamic components. Counter to our intuition, we find that the thermodynamic components do not always contribute to precipitation intensification. Although the thermodynamic temperature (\$pT\$ ) term, or CC scaling, strongly enhances EPS, especially in mid-to-high latitudes, the lapse rate term (\$pLR\$ ) and the pressure component (\$pPR\$ ) can weaken EPS. Specifically, the \$pLR\$ term not only correlates with saturation specific humidity, but is also affected by atmospheric stability and convective³⁸. However, these sub-grid processes cannot be well-captured by the convection parameterizations in GCMs, which may result in an underestimation of this term³⁹. In addition, we find that the dynamic component varies across different spatial-temporal scales, ranging from negative scaling to more than double CC scaling. We understand that the scaling behaviours cannot be directly applied to predict future precipitation extremes, nor can they be simply extrapolated to project long-term changes in extreme precipitation. However, detailed decomposition of this scaling and unravelling its future shifts could help to bound uncertainties in future extreme events and assess how their frequency and intensity will change.

38. Meredith, E. P., Ulbrich, U. & Rust, H. W. The Diurnal Nature of Future Extreme Precipitation Intensification. *Geophys. Res. Lett.* **46**, 7680-7689, (2019).

39. Gentine, P., Pritchard, M., Rasp, S., Reinaudi, G. & Yacalis, G. Could Machine Learning Break the Convection Parameterization Deadlock? *Geophys. Res. Lett.* **45**, 5742-5751, (2018).

5. On the choice of models: It is not entirely clear why the authors end up with these specific models, and only 1 (and 2 in one case) ensemble member for each model. Was this an issue of data unavailability? If not, then I see no particular reason why the authors should not extend their analysis to include more data.

Reply: Yes, at the current stage, only these 6 ensemble members are available for the

pressure data (e.g., vertical temperature and velocity profiles) at the Eday table scale in CMIP6. To better characterize the vertical structures of the atmosphere, here we employed the **Eday table outputs** (which contains 19 pressure levels) rather than the **day table outputs** (mostly only involves 8 pressure levels). This Eday table is a step forward in CMIP6 and was previously unavailable in CMIP5, and it is what enabled us to perform this study.

To address your concerns, we also downloaded CMIP5 model outputs (see Supplementary Table S2) and found similar results (except for smaller increases in T_{pp} under the future period). Further details can be found in response to **Reviewer #1 Q3**:

Supplementary Table 2. Basic information about the available CMIP5 model outputs

NO	Institution	Model name	Ensemble	Resolution Lon.×Lat.×Plev.
1	Canadian Centre for Climate Modelling and Analysis	CanESM2	r1i1p1f1	128×64×8
2	Centro Euro-Mediterraneo sui Cambiamenti Climatici	CMCC-CESM	r1i1p1f1	96×48×11
3	NOAA Geophysical Fluid Dynamics Laboratory	GFDL-CM3	r1i1p1f1	144×90×8
4	Japan agency for marine-earth science and technology	MIROC5	r1i1p1f1	180×90×8
5	Institut Pierre Simon Laplace	IPSL-CM5A-MR	r1i1p1f1	180×90×8
6	Max Planck Institute for Meteorology	MPI-ESM-MR	r1i1p1f1	192×96×15

Supplementary Figure 14. EPS, T_{pp} and dominant factor during the reference 1985-2014 period (a-c) and EPS anomalies, T_{pp} change and associated dominant factor during the future 2071-2100 future period relative reference period (d-f) based on the CMIP5 dataset. g, Zonal EPS and the DY , pPR , pT and pLR contributions to zonal EPS during the reference period. h, Zonal EPS anomalies and the associated DY , pPR , pT and pLR contributions to zonal EPS anomalies during the future 2071-2100 future period relative reference period.

6. On ERA5: To my knowledge rainfall is a model output in ERA5. If this is still the case in version 5, then the authors should clarify this in the Methods section, so that we make sure that the readers know this is not an “observational” dataset.

Reply: Sorry that this statement was misleading. We have rephrased our description of the ERA5 dataset in the revised Methods section as follows:

This ERA5 is based on the state-of-the-art Integrated Forecasting System (IFS) Cy41r2 and benefits from vast amounts of historical observations, developments in model physics, core dynamics and assimilation techniques, covering a period from 1950 to the present. Specifically, it is a model-based reanalysis product which has uncertainties of varying magnitude in different regions of the globe.

7. On the temporal resolution: Maybe the authors could consider performing the same analysis but at the hourly resolution (at least for ERA5 where hourly data are available).

It would be really interesting to see whether the contributions of the different components change at this resolution, and maybe this would help gain more insights on the results.

Reply: Thanks for this comment. We have added the decomposition results of **hourly** data from ERA5 to serve as a comparison. Furthermore, we additionally employ the **extended warm season** (May-October in the Northern Hemisphere and November-April in the Southern Hemisphere) data to avoid seasonal impacts on our results. All these experiments have been added in the revised **Decomposition of thermodynamic and dynamic contributions** Section as follows:

We also perform the decomposition at the hourly scale and for the extended warm season (May-October in the Northern Hemisphere and November-April in the Southern Hemisphere; Fig. 4). Based on the ERA5 dataset, we find the hook-like structure still governs global EPS over 68.4% of the globe at the hourly time scale and 59.4% during the warm season. The total scaling rates remain almost unchanged when comparing hourly and daily scales, and the results are robust when we focus on warm season in the mid-to-high latitudes ($>30^{\circ}\text{N}$ and $<30^{\circ}\text{S}$). This robustness is because the pT term dominates the total scaling over these regions and is stable regardless of different time scales (seasons). In contrast, in regions between $\sim 30^{\circ}\text{S}$ and $\sim 30^{\circ}\text{N}$ where the dynamic term prevails, the total scaling varies across different temporal scales and seasons. This partly reflects the high sensitivity of the dynamic term to rising temperature.

Fig. 4 | EPS, T_{pp} and dominant factor during the reference 1985-2014 period from ERA5 dataset. a,c,e Scaling rate, T_{pp} and dominant factor of the P_e-T relationship during the reference period at the hourly scale. **b,d,f** Scaling rate, T_{pp} and dominant factor of the P_e-T relationship during the reference period for the warm season (May to October in the Northern Hemisphere and November to April in the Southern Hemisphere). **g**, Zonal total scaling rate and the DY , pPR , pT and pLR contributions at the hourly scale. **h**, Zonal total scaling rate and the DY , pPR , pT and pLR contributions for the warm season.

Minor comments:

Title: Precipitation sensitivity refers to temperature, but it is not entirely clear in the title. Maybe consider changing that.

Reply: As the precipitation sensitivity is more common in wide-broader readership, we will keep the Title as “Large anomalies in future extreme precipitation sensitivity driven by atmospheric dynamics”.

Line 297: “The dynamic term plays”

Reply: Sorry for this typo and done.

Sincerely,

Dr. Yiannis Moustakis

Chair for physical geography and land use systems

Department of Geography

Ludwig-Maximilians Universität Munich

References:

- Moustakis, Y., Onof, C.J. & Paschalis, A. Atmospheric convection, dynamics and topography shape the scaling pattern of hourly rainfall extremes with temperature globally. *Commun Earth Environ* 1, 11(2020). <https://doi.org/10.1038/s43247-020-0003-0>
- O’Gorman, Paul A., and Tapio Schneider. “The physical basis for increases in precipitation extremes in simulations of 21st-century climate change.” *Proceedings of the National Academy of Sciences* 106, no. 35 (2009): 14773-14777.
- Wasko, C., and Sharma, A. (2014), *Quantile regression for investigating scaling of extreme precipitation with temperature*, *Water Resour. Res.*, 50, 3608– 3614, [doi:10.1002/2013WR015194](https://doi.org/10.1002/2013WR015194).

Reply: Thank you for providing these references; we have learnt and cited.

Reviewer #3 (Remarks to the Author):

The manuscript entitled “Large anomalies in future extreme precipitation sensitivity driven by atmospheric dynamics” aimed to decompose extreme precipitation sensitivity (EPS) to temperature into thermodynamic and dynamic components at the global scale. The main finding is that the lapse rate effect and the pressure component partly offsetting positive EPS. Future large EPS anomalies are caused by changes in updraft strength. However, authors misunderstood the relationship between long-term changes in precipitation extremes and warming climate and binning scaling of extreme precipitation. Many previous studies have pointed out they are different. I would recommend reject for the current version considering the high-quality requirement of Nature Communication.

Reply: We thank the reviewer for this comment to improve our manuscript. In this study, we intend to reveal the physics behind extreme precipitation scaling with rising temperature as well as its future shifting trajectories. As you mentioned, the binning scaling relationship cannot be directly applied to predict future precipitation extremes. In our manuscript, we did not use the binning scaling to project future long-term changes in precipitation under climate warming, but only focus on understanding the extreme precipitation sensitivity to temperature.

In fact, numerous studies have been devoted to explore the temperature scaling of precipitation extremes, and multiple approaches have been used, including linear regression based on annual maximum precipitation (Pfahl et al., 2017), quantile regression (Wasko et al., 2015; Moustakis et al., 2020) and the commonly employed binning scaling (Prein et al., 2017; Wang et al., 2017; Yin et al., 2018; Meredith et al., 2019; Zhang et al., 2022). To understand the underlying physical mechanism behind extreme precipitation sensitivity, we decompose the influencing contributions of extreme precipitation sensitivity into one dynamic and three thermodynamic components.

Pfahl, S., O’Gorman, P. A., & Fischer, E. M. (2017). Understanding the regional pattern of projected future changes in extreme precipitation. *Nature Climate Change*, 7(6), 423-427.

Wasko, C., Sharma, A., & Johnson, F. (2015). Does storm duration modulate the extreme precipitation-temperature scaling relationship?. *Geophysical Research Letters*, 42(20), 8783-8790.

Moustakis, Y., Onof, C. J., & Paschalis, A. (2020). Atmospheric convection, dynamics and topography shape the scaling pattern of hourly rainfall extremes with temperature globally. *Communications Earth & Environment*, 1(1), 11.

Prein, A. F., Rasmussen, R. M., Ikeda, K., Liu, C., Clark, M. P., & Holland, G. J. (2017). The future intensification of hourly precipitation extremes. *Nature climate change*, 7(1), 48-52.

Wang, G., Wang, D., Trenberth, K. E., Erfanian, A., Yu, M., Bosilovich, M. G., & Parr, D. T. (2017).

- The peak structure and future changes of the relationships between extreme precipitation and temperature. *Nature Climate Change*, 7(4), 268-274.
- Yin, J., Gentile, P., Zhou, S., Sullivan, S. C., Wang, R., Zhang, Y., & Guo, S. (2018). Large increase in global storm runoff extremes driven by climate and anthropogenic changes. *Nature communications*, 9(1), 4389.
- Meredith, E. P., Ulbrich, U., & Rust, H. W. (2019). The diurnal nature of future extreme precipitation intensification. *Geophysical Research Letters*, 46(13), 7680-7689.
- Zhang, S., Zhou, L., Zhang, L., Yang, Y., Wei, Z., Zhou, S., ... & Dai, Y. (2022). Reconciling disagreement on global river flood changes in a warming climate. *Nature Climate Change*, 1-8.

To fully consider your concerns, we have revised our manuscript from the following four perspectives:

- (1) We systematically validated the robustness of our main conclusions by applying the quantile regression approach, and we also evaluate our results at different temporal scales and seasons. Please refer to the revised Section Decomposition of thermodynamic and dynamic contributions and Fig. 4, Supplementary Fig. 2.
- (2) We have compared our CMIP6 results with the CMIP5-oriented simulations. Please refer to the revised Section Discussion and Supplementary Fig. 14.
- (3) We have fully discussed the synoptic patterns behind extreme precipitation by using various large- and local- scale meteorological variables. Please refer to the revised Section Discussion and Supplementary Figs. 12-13.
- (4) We have provided a brief discussion about the issue in the Discussion Section as follows:

Counter to our intuition, we find that the thermodynamic components do not always contribute to precipitation intensification. Although the thermodynamic temperature (pT) term, or CC scaling, strongly enhances EPS, especially in mid-to-high latitudes, the lapse rate term (pLR) and the pressure component (pPR) can weaken EPS. Specifically, the pLR term not only correlates with saturation specific humidity, but is also affected by atmospheric stability and convective³⁸. However, these sub-grid processes cannot be well-captured by the convection parameterizations in GCMs, which may result in an underestimation of this term³⁹. In addition, we find that the dynamic component varies across different spatial-temporal scales, ranging from

negative scaling to more than double CC scaling. We understand that the scaling behaviours cannot be directly applied to predict future precipitation extremes, nor can they be simply extrapolated to project long-term changes in extreme precipitation. However, detailed decomposition of this scaling and unravelling its future shifts could help to bound uncertainties in future extreme events and assess how their frequency and intensity will change.

We sincerely hope this explanation could satisfy the reviewer's concerns and the revised version to be published this time.

1. In the beginning of Introduction and abstract, authors emphasize the intensification of precipitation extremes under climate warming, the extreme precipitation sensitivity (EPS) to temperature should be the relationship of long-term changes in annual precipitation extremes with warming climate. However, authors established the precipitation-temperature (P-T) relationship based on their intra-annual variation, which is referred to the binning scaling. And the following results are about discussion the binning scaling. However, there is fundamental methodological distinction between the relationship of long-term changes in annual precipitation extremes with warming climate and binning scaling. Zhang et al., (2017, <https://doi.org/10.1038/ngeo2911>) and Sun et al., (2020, <https://doi.org/10.1175/JCLI-D-19-0920.1>) have discussed their differences from statistical and physical interpretations. Many previous studies, such as Bao, J. et al. 2017 (<https://doi.org/10.1038/nclimate3201>), Prein et al., 2017 (<https://doi.org/10.1038/nclimate3168>) have pointed out the relationship of extreme precipitation and climate warming are different from the hook-shaped scaling. Binning curves (like the hook-shaped scaling the study mentioned) reflect seasonal changes in the relationship between daily temperature and extreme precipitation.

Reply: We notice that the binning scaling method is different from predicting future precipitation extremes using warming temperature. To further evaluate our main conclusions, we have provided sensitivity analyses with different methods, temporal scales and more climate ensemble members.

All these supplementary experiments indicate that the existence of the hook-shaped scaling is reasonable, just as previous studies reported; and **binning curves do not reflect seasonal changes at the daily scale but reveal long-term changes in extreme precipitation with warming environment**. Details can be found in the above Responses and figures as follows:

Supplementary Figure 2. EPS and T_{pp} based on the quantile regression and the Akaike information criterion during the reference 1985-2014 period using ERA5 dataset.

Fig. 4 | EPS, T_{pp} and dominant factor during the reference 1985-2014 period from ERA5 dataset. **a,c,e** Scaling rate, T_{pp} and dominant factor of the P_e - T relationship during the reference period at the hourly scale. **b,d,f** Scaling rate, T_{pp} and dominant factor of the P_e - T relationship during the reference period for the warm season (May to October in the Northern Hemisphere and November to April in the Southern Hemisphere). **g**, Zonal total scaling rate and the DY , pPR , pT and pLR contributions at the hourly scale. **h**, Zonal total scaling rate and the DY , pPR , pT and pLR contributions for the warm season.

Supplementary Figure 14. EPS, T_{pp} and dominant factor during the reference 1985-2014 period (a-c) and EPS anomalies, T_{pp} change and associated dominant factor during the future 2071-2100 future period relative reference period (d-f) based on the CMIP5 dataset. g, Zonal EPS and the DY , pPR , pT and pLR contributions to zonal EPS during the reference period. h, Zonal EPS anomalies and the associated DY , pPR , pT and pLR contributions to zonal EPS anomalies during the future 2071-2100 future period relative reference period.

2. Also, this study cited many previous studies (such as Pfahl, S., et al., 2017; O’Gorman and Schneider, 2009) to help explain the method or results, while these studies mainly focus on the relationship of long-term changes in annual precipitation extremes with warming climate, but not the relationship between daily temperature and daily precipitation.

Reply: Thanks for this comment. In this study, what we focus on is the **relationship between extreme precipitation and local temperature** at the daily scale, rather than relationship between daily precipitation and daily temperature. We intend to explore the shifting mechanisms behind the extreme precipitation under climate warming. To achieve this, we creatively decompose the extreme precipitation scaling with temperature into four components (including three thermodynamic terms and one dynamic term) and we find that the thermodynamic components do not always contribute to precipitation intensification. Further, the dynamic term is highly sensitive

to future warming and could lead to large uncertainty in future extreme precipitation. To validate our conclusion, we employ different approach (e.g., quantile regression and the Akaike information criterion), different models (e.g., CMIP5 ensembles) and perform study at different temporal scale (e.g., hourly scale) and at different season (e.g., only focus on the warm season). All these supplementary experiments have validated the robustness of our conclusion.

Reviewer #1 (Remarks to the Author):

After reviewing the revised manuscript, I now come to the conclusion that the authors have worked very thoroughly and conscientiously and I therefore see no further points of criticism. In my opinion, the concerns of reviewer 3 were also dealt with confident and counterarguments were given.

Reviewer #2 (Remarks to the Author):

Dear Editor,

The authors have carefully addressed all of my comments. I can hence recommend this manuscript for publication.

Sincerely,
Dr. Yiannis Moustakis